# Extra virgin olive oil mitigates lung injury in necrotizing enterocolitis: Effects on TGFβ1, Caspase-3, and MDA in a neonatal rat model

**Mustafa Tuşat**[1]*, **Recep Eröz**[2], **Ferhan Bölükbaş**[3], **Erkan Özkan**[4],
**Mehmet Semih Demirtaş**[5], **Hüseyin Erdal**[3], **Osman Okan Özocak**[6]

**1** Aksaray University Medical Faculty, Department of Pediatric Surgery, Aksaray University, Aksaray, Turkey, **2** Aksaray University Medical Faculty, Department of Medical Genetics, Aksaray University, Aksaray, Turkey, **3** Aksaray University Veterinary Faculty, Department of Histology and Embryology, Aksaray University, Aksaray, Turkey, **4** Aksaray University Veterinary Faculty, Department of Parasitology, Aksaray University, Aksaray, Turkey, **5** Aksaray University Medical Faculty, Department of Pediatrics, Aksaray University, Aksaray, Turkey, **6** Erciyes University Medical Faculty, Department of Cardiovascular Surgery, Erciyes University, Kayseri, Turkey

\* mustafatusat42@hotmail.com

## Abstract

### Background

Necrotizing enterocolitis (NE), which is common in premature babies, has been associated with lung damage. Our aim is to explore the effect of enterally administered extra virgin olive oil (EO) with rich polyphenol content on clinical parameters, histopathological score, *Transforming growt factor beta-1* (*TGFβ1*), Caspase 3 and Malondialdehyde (MDA) levels in NE-related lung injury of neonatal rats.

### Methods

Three groups (control, NE, NE+EO) were created, with 8 neonatal rats in each group. NE was induced by hypoxia-hyperoxia-hypothermia and formula feeding. EO was given to the treatment group by orogastric probe for 3 days. Intestinal and lung tissue were excised for analysis.

### Results

*TGFβ1* expression levels, TGFβ1 and MDA concentration levels were higher in the NE compared to NE + EO and control groups (p < 0.001), and their levels decreased after EO treatment compared to the NE group (p < 0.001). It was determined that EO treatment significantly reduced the histopathological damage and the caspase-3 (CASP3) expression level in the lung (p < 0.001).

### Conclusion

Our findings emphasize that *TGFβ1* has an crucial function in NE-related lung injury and that EO has therapeutic potential in NE-related lung injury.

**Data availability statement:** All data underlying the findings described in the article are available at URL passcode ''https://osf.io/6za8q/'' and DOI number ''' DOI 10.17605/OSF.IO/6ZA8Q'' in a public repository. All data were also uploaded to PLOS ONE as supporting information in the form of an excel file.

**Funding:** The author(s) received no specific funding for this work.

**Competing interests:** The authors have declared that no competing interests exist.

## 1. Introduction

Necrotizing enterocolitis (NE) is a life-threatening gastrointestinal problem characterized by inflammation, necrosis and frequently affecting the terminal ileum and colon, seen especially in premature newborns in the presence of various risk and etiological factors [1]. It is known that NE affects many organs [2] and lung damage occurs in 10–15% of premature babies with NE [3]. This lung damage is an important complication of NE, and NE is an important predisposing risk factor for the development of more severe lung damage than broncho pulmonary dysplasia (BPD), which can often be observed in premature babies [2,4].

*Transforming growth factor-beta-1* (*TGFβ1*) is a member of the TGF-β family of growth factors, which has various physiological roles such as apoptosis, inflammation, proliferation, differentiation, and control of immune function, and *TGFβ1* is the *TGF* isoform most frequently formed in tissue damage [5]. *TGFβ1* exerts its effects through the *small mother against decapentaplegic (SMAD)/TGFβ1* cascade. SMAD proteins are nuclear proteins in *TGF-β* signaling. In *SMAD/TGFβ1* signaling, TGFß1 first binds to TGF-ß receptor (TßR)1 and 2 receptors and then TßR-1 phosphorylates SMAD2 and SMAD3 proteins, enabling the activation of these structures. Activated SMAD2 and SMAD3 proteins form a complex with SMAD4 and translocate to the nucleus to regulate transcription in genes where *TGFβ1* is effective [6].

Caspase-3 protein is a member of the cysteine-aspartic acid protease (caspase) family and is encoded by the *cysteinyl aspartate-specific proteinase-3* (CASP3) gene (located on the long arm of Chromosome 4 (region 4q35.1) and contains nine exons) is a common key enzyme of the execution phase of cell apoptosis and pyroptosis pathways [7]. Caspases are synthesized in the pro-caspase form, i.e., as inactive zymogens, and are activated upon appropriate stimulation. In response to apoptotic signaling events, CASP3 is cleaved and activated by upstream initiator caspases (caspase-8,10) and transported to the nucleus to cleave nuclear proteins, resulting in apoptotic nuclear changes [7,8].

It is known that *TGFβ1* plays a role in lung damage in which sepsis, fibrosis, inflammation and epithelial-mesenchymal transformation (EMT) occur [9,10]. It has been reported that pro-inflammatory *TGFβ1* cytokine levels are increased in the lungs of babies with neonatal hyaline membrane disease and neonatal BPD [11] and that the expression and protein levels of proinflammatory *TGFβ1* increase in the bronchoalveolar-lavage fluid (BALF), lung tissue in acute lung injury (ALI) and bleomycin-induced lung fibrosis [12,13].

Extra virgin olive oil (EO), fundamental component of the Mediterranean diet, has positive effects on human health mainly caused by the polyphenol content of EO [14]. In studies conducted on humans and animals in vivo or in vitro, EO has beneficial effects resulting from its anti-inflammatory, antioxidant and antimicrobial properties [1,14,15]. Enteral administration of tyrosol (Tyr), Oleuropein (Ole) and hydroxytyrosol (HTyr), which are polyphenol compounds of EO, reduces lipopolysaccharide (LPS)-induced ALI by inhibiting inflammation and oxidative stress (OS) [16–19]. The unsaponifiable fractions of orally administered EO reduce aluminum chloride ($AlCl_3$) and acrylamide-induced lung and DNA damage [20], and Ole has an antifibrotic effect by exhibiting anti-inflammatory and antioxidant properties in bleomycin-induced lung fibrosis in rats [21]. In a study investigating the effect of orally administered 25% HTyr containing olive leaf extract on rats with the acetic acid-induced colitis model, it was shown that HTyr inhibited OS and inflammation by reducing colonic MDA and *TGFB1* expression [22]. In dextran sodium sulfate-induced colitis models, it was determined that the use of EO caused a significant decrease in the expression levels of *TGFB1* and *CASP3* in colon tissue and alleviated inflammation [23,24]. It has been reported that intraperitoneal and oral administration of olive oil significantly reduced the levels of MDA and TGFβ1 in liver tissue in mice with carbon tetrachloride ($CCl_4$)-induced liver fibrosis [25–27]. It has

been reported that Tyr caused significant decreases in MDA and *CASP3* expression levels in rats with testicular damage caused by $AlCl_3$ [28]. In vitro renal hypoxia model induced by cobalt chloride ($CoCl_2$) using human renal proximal tubular cells (HK-2), it was observed that the application of HTyr caused a significant decrease in the *TGFB1* gene expression levels, which is effective in inflammation and fibrosis [29]. It was detected that oral administration of EO rich in triterpenic acids for 8 weeks caused a significant decrease in *TGFB1* level and reduced collagen accumulation in the arterial walls of hypertensive rats [30]. A strong correlation between serum MDA and placental TGFB1 has been shown in preeclamptic rats, and researchers reported that EO treatment reduced abnormally elevated levels of MDA and TGFB1 in placental tissue [31]. Another study reported that treatment with Ole reduced CASP3 expression in brain tissue in a mouse model of hypoxic-ischemic encephalopathy [32]. It has been reported that phenolics secoiridoids such as HTyr, Ole aglycone in EO, act as anti-aging phytochemicals by inhibiting *TGFB1*-induced fibrogenic and oncogenic EMT in Madin–Darby canine kidney (MDCK) cells and human breast cancer (MCF-7) cells [33]. In addition, in our study investigating the protective effect of EO on the intestines of rats with experimental NE, we detected that EO treatment exhibited anti-inflammatory, antioxidant, anti-apoptotic and cytoprotective effects by significantly reducing the levels of MDA, interleukin 1 beta (IL1β), interleukin 6 (IL6), Epidermal growth factor (EGF) and the number of CASP3 positive cells (Casp-3⁺ cells) [1]. Considering the anti-inflammatory, antioxidant and antimicrobial properties of EO and related studies, it is likely that EO has a positive effect on NE-related lung damage. To our knowledge, the effect of EO rich in phenol content on lung damage in rats with NE has not been investigated.

In our study, we aimed to evaluate the impact of enterally administered EO with rich polyphenol content on clinical parameters, histopathological scoring (HPS), Casp-3⁺ cells, *TGFβ1* gene expression level and both TGFβ1, MDA lung tissue homogenate level on lung damage caused by the NE model experimentally created in rats.

## 2. Materials and methods

### 2.1 Ethics

For this study, approval was received from the Animal Experiments Ethics Committee of Aksaray University Experimental Animals Application and Research Center (dated 24/04/2024 and decision number 2024/04–16) and the National Institute of Health's Guide for the Care and Use of Laboratory Animals was followed throughout the experiment and all procedures are reported in accordance with ARRIVE guidelines. Neonatal rats were sacrificed with high doses of ketamine and xylazine before surgical procedures. Every effort was made to minimize pain during the experimental procedures.

### 2.2 Animals

Neonatal rats used in the study were obtained from Aksaray University Animal Experiments Application and Research Center. For the study, 4 Wistar Albino female rats, 10–12 weeks old and weighing 200–250 g, were impregnated and kept with in a 12-hour light/dark environment, at 23 ± 1 ˚C temperature and 55–60% humidity in polycarbonate cages. Pellet feed and drinking water were provided ad libitum. Starting from the 20th day of pregnancy, the rats were transferred to cages and monitored. As birth approaches, the frequency of observation was increased. Births were carried out under the supervision of a veterinarian, and neonatal rats were randomly divided into 3 groups, with 8 members. To avoid the protective effect of breast milk, neonatal rats to be induced NE were taken from their mothers immediately after birth and placed in an incubator with a temperature of 37 °C and 65–70% humidity, 12 hours

dark and 12 hours light. The health and behavior of neonatal rats were monitored hourly during the experiment. Our research team included a veterinarian and the other researchers were certified in the handling of experimental animals. Induction of NE was performed for 3 days and rats were sacrificed on the morning of day 4 performed by intraperitoneal administration of 10 mg/kg xylazine and 90 mg/kg ketamine [34]. Acetaminophen was prepared to be administered oragastrically at a dose of 200mg/kg in the event that pain signs were observed in rats [35].

## 2.3 Experimental model

Neonatal rats in group 1 (Control) stayed with the mother and were fed with breast milk, and this group was not exposed to any procedures. NE induction to the rats in group 2 (NE) and group 3 (NE+EO) was performed according to the model specified by Güven et al. [36]. For NE induction, neonatal rats were fed with formula consisting of a mixture of 100 cc puppy milk (Baephar-Bogena. BV Sedel. The Netherlands) and 20 g similac 60/40 (Ross-Pediatrics. Columbus. Ohio) by gavage and 0.2 ml each time. The daily dose was increased by 0.1ml according to tolerance. Additionally, neonatal rats were exposed to 100% Carbon dioxide ($CO_2$) inhalation for 10 minutes, followed by 97% Oxygen ($O_2$) inhalation for 5 minutes, and cold for 5 minutes at +4°C, twice a day for 3 days [36]. EO, which contains rich polyphenols, was given to neonatal rats in Group 3 at a dose of 2ml/kg via oral gavage for 3 days. Baby rats in control and NE groups were given distilled water at a dose of 2ml/kg via oral gavage. EO (Sidyma, Muğla, Turkey), which contains rich polyphenols with a polyphenolic content of 832mg/kg, was purchased from the local market and the polyphenolic compound levels and fatty acid levels are stated in Table 1.

## 2.4 Humane endpoints of work

In order to prevent unnecessary pain, suffering and stress in animals used in research, which are both ethically and scientifically important, the humane endpoint criteria specified in the studies were complied with in our study. The Humane endpoints in our study address the physical and psychological status of the animals, such as general health, behavioral changes,

**Table 1. Content of EO administered to rats.**

| Fatty Acid | (%) | Phenolic Composition | ppm |
|---|---|---|---|
| Oleicacid | 71.07 | Oleuropeinaglycon (monoaldehyde form) | 58 |
| Palmiticacid | 14.32 | Oleuropeinaglycon (dialdehyde form) | 78 |
| Linoleicacid | 9.08 | Oleocantral + Oleacein (index D1) | 469 |
| Steraicacid | 2.36 | Ligstrosideaglycon (monoaldehyde form) | 55 |
| Linolenicacid | 0.98 | Oleocantral | 321 |
| Palmitoleicacid | 0.83 | Ligstrosideaglycon (dialdehyde form) | 172 |
| Arachidicacid | 0.45 | Oleacein | 148 |
| Heptadecenoicacid | 0.19 | Hydroxytyrosol (mg/kg) | 283 |
| Beheric-acid | 0.12 | Tyrosol | 549 |
| Gadoleic-acid | 0.34 | **Total amount of phenolic compounds** | **832** |
| Lignoseric-acid | 0.11 | | |
| Heptadecanoic-acid | 0.12 | | |
| Alfa-Tocopherol (ppm) | 605 | | |

**ppm:** parts per million.

Fatty Acid Percentage, Phenolic and Polyphenolic Composition of the EO Administered to Rats.

pain, weight loss, loss of appetite, and respiratory distress [37]. In our study, if the threshold values of these criteria were reached, euthanasia was performed with intraperitoneal administration of 10 mg/kg xylazine and 90 mg/kg ketamine [34]. The humanitarian endpoint criteria established in the study were evaluated. These criteria are

1  -When severe stress is observed in animals (absence of the sucking reflex)

2  -If the rat is in severe pain or suffering (continuous struggle)

3  -Significant weight loss observed during the course of the experiment, especially if the animal's weight drops by more than 20%.

4  -When prolonged severe loss of appetite is observed.

5  -Complete cessation of movement in the neonatal rat.

6  -Graying of the whole body due to severe breathing difficulties.

7  -Severe impairment of heart rate, body temperature, respiratory rate and reflexes

In the present study, humane endpoints were tabulated and neonatal rats were monitored hourly throughout the experiment. On the 3rd day of the study, loss of movement and severe respiratory distress were observed in an animal in the NE group. Since the humane endpoint criterion was reached before the end of the experimental period, the neonatal rat was euthanized with high dose anesthesia 15 minutes after the determination of the humane endpoint criterion under the control of a veterinarian. In our study, there were no rats that died without meeting the euthanasia criteria. The other 23 neonatal rats used in our study completed the experiment without reaching the human endpoint criteria.

## 2.5  Clinical illness score (CIS) and follow-up

Neonatal rats were weighed every day throughout the experiment with a very sensitive balance, including birth weight (BW) and last day weight (LW). The daily CIS of neonatal rats was evaluated according to the scoring system determined by Zani et al [38]. For this purpose, a blind observer calculated a CIS. Deaths due to NE and other causes were recorded daily throughout the experiment.

## 2.6  Macroscopic evalution

On the morning of day 4 of the experimental model, neonatal rats were sacrificed with a high dose of anesthetic drug. After entering the abdomen through a median incision and the thorax through a sternotomy, the abdomen and chest cavity were explored. Macroscopic evaluation was performed according to the method described by Zani et al. and bowel macroscopic score (BMS) was calculated [38]. Then, 3 cm of intestine was excised, including the terminal ileum and proximal colon. In the lung tissue sample, the lower lobe of the right lung was excised. Half of the samples were reserved for immunohistochemical and histopathological analysis, while the other half were washed with saline and stored at −80 °C for genetic and biochemical investigations

## 2.7  Detection of TGFβ1 and MDA levels via enzyme-linked immunosorbent assay (ELISA) in lung tissues

The tissues were rinsed with pre-cooled PBS to completely remove excess blood before homogenization. Then, an equal weight of tissue (~70 mg) from each group was weighed using a precision scale. Following, 1ml of 1X pre-cooled PBS was added on the tissues and

homogenized via bead mill homogenizer (Scientific industries, digital disruptor genie Cat no:si-dd38) at 2000 rpm for 5 minutes. Finally, the supernatant were collected and stored in aliquots at ≤ −20 °C for ELISA.

Lung tissue samples from whole rats were analyzed in duplicate. TGFβ1 lung tissue levels were analysed by ELISA method with Rat *TGFβ1* ELISA Kit [catalog number: ELK2311]. The analysis results were expressed as nanograms/milliliter (ng/mL). The MDA lung tissue levels were analysed with ELISA method and ELK Biotechnology Rat MDA ELISA Kit [catalog number: ELK8612].

## 2.8  Histopathologic evalution of intestine

Tissue specimens were fixed in 10% formalin and embedded in paraffin blocks. Sections (6 μm thick) were taken from the blocks and stained with the Hematoxylin-eosin [39] method for histopathological examination. The prepared preparations were examined under a microscope (Leica DM2500, Leica Microsystems GmbH, Germany) and photographs of the necessary areas were taken with a digital camera (Leica DFC 320). Changes observed as a result of histopathological examination of preparations stained with hematoxylin-eosin were scored between 0 and 4. Scoring was performed according to the method described by Caplan 1994 [40]. NE is indicated by a score of 2 or higher, while severe NE is indicated by a score of 3 or higher.

## 2.9  Histopathologic evalution of lung

Lung tissue specimens taken from newborn rats were fixed in 10% formalin for 24 hr, underwent a series of histological preparations and, were sectioned into 6 μm thick tissue sections using a rotary microtome. The sections were stained with Hematoxylin-eosin staining method [39]. Histopathological evaluations were performed using a modified version of the scale developed by the American Thoracic Society (ATS), Made according to Drucker et.al. 2018 (Table 2). Scores from 0 (normal) to 2 (acute lugn injury) are given for six different parameters and added up to obtain a score between 0 and 12; 0 indicates normal lung, and 12 indicates severe acute injury with bleeding [41,42].

## 2.10  Immunohistochemical evalution of lung

Staining of tissue samples taken for immunohistochemical procedures was carried out with a procedure based on streptavidin-biotin-peroxidase complex (sABC). For this purpose, 6 μm thick sections taken on poly L-lysine coated slides were dried by keeping them in an oven at 37°C overnight. Subsequently, the sections were deparaffinized and rehydrated and boiled in a

**Table 2.  Lung injury scoring system.**

| Parameters | Score per field X400 magnification | | |
|---|---|---|---|
| | 0 | 1 | 2 |
| Neutrophils in the alveolar space | None | 1–5 | >5 |
| Neutrophils in the interstitial space | None | 1–5 | >5 |
| Hyaline membranes | None | 1 | >1 |
| Proteinaceous debris filling the airspaces | None | 1 | >1 |
| Alveolar septal thickening | <2× | 2×–4× | >4× |
| Red blood cells in alveolar space | Few | Half-filled | Filled |

Lung injury score = sum of all scores, range from 0 to 12.

microwave oven in citrate buffer solution (pH 6) for 5 minutes for antigen retrieval. To inhibit endogenous peroxidase activity, the sections were kept in 3% hydrogen peroxide solution for 20 minutes and nonspecific binding sites were blocked by incubating in blocking solution (Thermo Fisher Scientific Inc., UK) for 5 minutes. The sections were then incubated with the primary antibody (Caspase-3 (STJ97448, St John's Laboratory, 1:200 dilution) for 1 hour at room temperature, followed by biotinylated goat polyvalent antibody (Thermo Fisher Scientific Inc., UK) for 30 minutes. Sections washed with buffered phosphate saline (PBS, Biotech) were incubated with streptavidin-peroxidase (HRP, Thermo Fisher Scientific Inc., UK) for 30 min at room temperature. While DAB (3–3'-diaminobenzidine, Thermo Fisher Scientific Inc., UK) was used as chromogen, Mayer's hematoxylin solution was preferred for nuclear staining. Negative control preparations were prepared by incubating tissue sections with PBS instead of primary antibody. The stained sections were covered with coverslips using synthetic adhesive (Entellan, Merck) after being dehydrated through a graded alcohol series and passed through xylene.

All preparations were examined under a light microscope (Leica DM2500) and then were photographed by a digital camera (Leica DFC 320). In the sections stained with the immuno-histochemical method, all Casp-3$^+$ cells were counted in 6 different randomly selected regions of 10.000 μm$^2$ at x400 magnification. All evaluations were performed by two researchers blinded to the sample identification.

## 2.11 Ribonucleic Acid (RNA) isolation and complementary Deoxyribonucleic Acid (cDNA) synthesis

RNA was isolated from lung samples via both Hybrid-R (Catalog No: 305–101) and RiboEx (Catalog No: 301–001) isolation kits base on the manufacturer's protocol. cDNA, copy of the RNA molecule, was obtained using A.B.T.™ cDNA Synthesis Kit with Rnase Inh. (High Capacity) cDNA synthesis kit (Catalog No: C03–01–05) via ProFlex thermalcycler. A total 20 μL mastermix (10 μl isolated RNA sample, 0.5 μl RNase Inhibitor, 2 μl Random hexamer, 2 μl 10X Reaction Buffer, 1 μl 20X dNTP mix, 1 μl RTase, 3.5 μl RNase free water) was used for cDNA synthesis. cDNA synthesis condition were at 25 °C for 10 m in step 1, at 37 °C for 120 m in step 2, at 85 °C for 5 m in step 3 and finally at 4 °C for ~ m.

## 2.12 *TGFβ1* and *glyceraldehyde-3-phosphate dehydrogenase* (*GAPDH*) gene expressions by real-time quantitative polymerase chain reaction (qPCR)

The following primers were used for the genes expression

| Gene Name | Primers | Manufacturer |
|---|---|---|
| *TGFβ1*-Forward | AGGACCCAGATACTCCCAAG | Oligomer |
| *TGFβ1*-Reverse | GTCCCCATACACTGCTTCAC | Oligomer |
| *GAPDH*-Forward | GTGGAGTCCTGGAACTGAAGC | Oligomer |
| *GAPDH*-Reverse | AGCACCAATCTGTGATGACAAC | Oligomer |

For each cDNA sample, expression of *TGF1β* and the reference gene (*GAPDH*) were detected via the Applied Biosystems™ QuantStudio5 Real-Time PCR System. Polymerase chain reactions (PCRs)were performed using Mastermix that include 4 μl cDNA Template, 10 μl 2X MasterMix (with SYBR-Green), 1 μl Forward Primer (10 μM), 1 μl Reverse Primer (10 μM), 3 μl RNase-Free Distilled Water, 1 μl ROX Dye with a final volume of 20 μL.

Cycle conditions of PCR were 1 cycle initial denaturation at 95 °C for 300 sec, 40 cycles denaturation at 95 °C for 15 s and 40 cycles annealing at 60 °C for 60 second. The *GAPDH* was used as reference gene and fold Change had been calculated.

## 2.13  Statistical analysis

The Statistical Package for Social Sciences (IBM Corp., Armonk, NY, USA) 22.0 were used for the data analyze (S1 Dataset). Resource Equality Method, which is a frequently used method, based on the Degrees of Freedom error was used to determine the number of subjects to be assigned to groups. The distribution of data was detected via Shapiro Wilk test. The descriptive statistic was done. Because the normally distribution of data ($p > 0.05$), One-Way Anova test was used for each group comparisons. Since the variances of the groups were homogeneous, Tukey HSD test was used. Also polynominal regression test was carried out. The $p < 0.05$ was accepted as statistically significant.

## 3.  Results

On the 3rd day of the study, one rat in the NE group showed signs of human endpoint and the neonatal rat was euthanized and subsequent laparotomy revealed necrosis and perforation in the terminal ileum. There were meaningful differences among the groups for LW, BMS, CIS, *TGFβ1 expression,* MDA and TGFβ1 concentration levels ($p < 0.05$) (Table 3, Fig 1).

HPS of bowels, Casp-3⁺ cells and HPS for lungs were significantly difference among the goups ($p < 0.001$) (Table 4).

The lungs of rats in control group showed normal histology. In NE group, significant histopathological lesions such as neutrophil infiltration, hyperemia, bleeding, and thickening of the interalvelor septum were detected in the lung tissue. It was observed that the histopathological damage to the lung was importantly reduced in the NE+EO group ($p < 0.001$) (Tables 4, 5 Fig 2A, 2B, 2C S1 Fig, S2 Fig, S3 Fig).

Immunohistochemically, CASP3 activity was evaluated to detect the apoptosis process in the lung in all groups. It was determined that CASP3 expressions increased in the lung cells of the NE group. It was observed that CASP3 expressions decreased significantly in the NE+EO group ($p < 0.001$) (Table 4, 5 Fig 3B, 3C, 3D S4 Fig, S5 Fig, S6 Fig, S7 Fig).

According to binary comparison, important differences between NE and NE+EO, between control and both NE and NE+EO were detected in terms of LW, CIS and BMS ($p \leq 0.001$), (Table 5).

Also meaningful differences between NE and NE+EO, between control and NE were detected in terms of *TGFβ1* Expresssion Levels, Concentration Levels of TGFβ1 and MDA ($p < 0.001$), (Table 5).

**Table 3.  The mean concentration and expression levels of *TGFβ1*, MDA and clinical parameters of all groups.**

| Parameters | Groups | | | |
|---|---|---|---|---|
| | Control (M ± SD)(min-max) | NE (M ± SD) (min-max) | NE+EO(M ± SD) (min-max) | p |
| BW | 6.125 ± 0.58 (5.2–7.2) | 6.075 ± 0.689 (5.4–7.1) | 6.05 ± 0.316 (5.7–6.6) | 0.963 |
| LW | 8.85 ± 0.52 (8.2–9.7) | 6.325 ± 0.643 (5.6–7.2) | 7.70 ± 0.363 (7.4–8.5) | **0.000*** |
| CIS | – | 10.13 ± 1.458 (8–12) | 2.38 ± 1.302 (0–4) | **0.000*** |
| BMS | – | 5.13 ± 0.641(4–6) | 1.63 ± 0.518 (1–2) | **0.000*** |
| *TGFβ1* C. Lev | 166.11 ± 22.89 (128.3–198.6) | 474.89 ± 94.85(335.83–585.25) | 186.65 ± 20.101 (150.19–205.93) | **0.000*** |
| *MDA* C. Lev | 110.39 ± 8.22 (103.68–127.26) | 216.73 ± 33.70 (183.14–272.72) | 114.158 ± 5.184 (105.45–119.31) | **0.000*** |
| *TGFβ1* Exp Lev | 0.558 ± 0.656 (0.08–1.91) | 9.883 ± 2.99 (4.8–14.75) | 2.648 ± 3.319 (0.15–8.49) | **0.000*** |

*NE*: Necrotizing Enterocolitis *EO*: Extra Virgin Olive Oil **BW**: Birth Weight **LW**: Last Day Weight **CIS**: Clinical Illness Score **BMS**: Bowel Macroscopy Score *TGFβ1*:Transforming growth factor beta 1 *MDA*: Malondialdehyde **C**: Concentrations **Exp**: Expression **Lev**:Levels *M*: Mean, *SD*: Standard Deviation **min**: minimum **max:** maximum

*= Statistically significant.

When the Casp-3⁺ cells and HPS for lungs to be considered, statistically significant differences between NE and NE+EO, between control and both NE and NE+EO were detected ($p \leq 0.001$), (Table 5).

According to polynominal regression analaysis results, meaningful relations were detected between TGFβ1 Concentration levels and all of LW, CIS, BMS, MDA, *TGFβ1* Expressions, Casp-3⁺ cells and HPS for lung ($p < 0.001$) (Table 6, Fig 4).

Additionally, there were meaningful relations between MDA and all of LW, CIS, BMS, *TGFβ1* Expressions, Casp-3⁺ cells and HPS for lung ($p < 0.001$) (Table 6, Fig 5).

When TGFβ1 expression levels to be considered, meaningful relation between *TGFβ1* expression levels and all of LW, CIS, BMS, MDA, Casp-3⁺ cells, HPS and TGFβ1 Concentration levels detected for lung ($p < 0.001$) (Table 7, Fig 6).

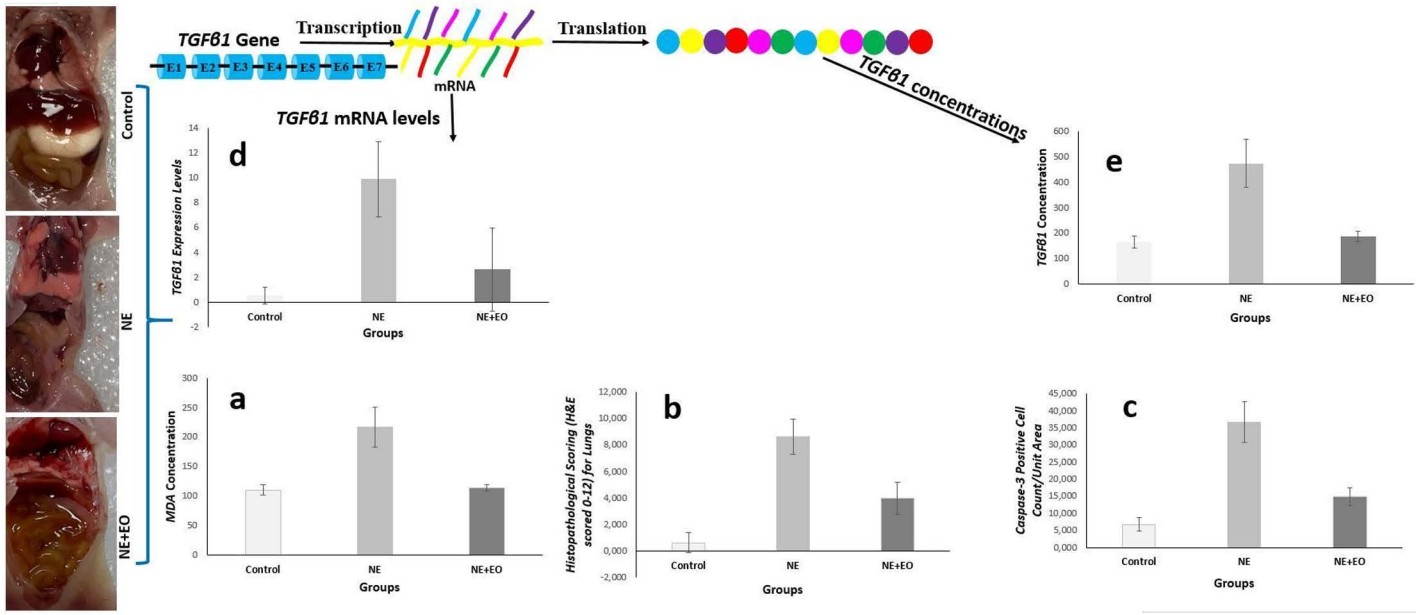

**Fig 1. Illustrative examples of our study.** Three main groups rats as control, NE, and NE+EO each of them with 8 members were included in the study. Both *TGFβ1*gene expression and concentration levels, MDA concentration levels, histopathological scoring and Casp-3⁺ cells count/unit area of groups were detected. As a response to the cellular damage caused by NE, the expression and translation levels of the different genes may be changed in the nucleus of cells. Depending on the NE, the lung tissue concentration level of MDA significantly increased (a). Also the histopathological scoring (H&E scored 0–12) (b) and Casp-3⁺ cells count/unit area (c) in lung tissue significantly increased depending on the NE injuries. The *TGFβ1*gene is transcribed *TGFβ1* mRNA, and *TGFβ1* proteins are translated from mature *TGFβ1* mRNA. There was statistically significant differences among the groups in terms of *TGFβ1 gene* expression levels. The expression level of *TGFβ1* gene significantly increased depending on the NE injuries (d). Also depending on the NE, the lung tissue concentration level of *TGFβ1* significantly increased (e). It may be said that, the EO has an important protective effect against lung tissue damage caused by NE injuries.

**Table 4. Casp-3⁺ Cells Count/Unit Area and HPS of all groups.**

|  | Control MD (25–75) | NE MD (25–75) | NE+EO MD (25–75) | p |
|---|---|---|---|---|
| **HPS for lungs** | 0.5 (0–1) | 9 (7.25–9) | 3.5 (3–5) | **0.000*** |
| **Casp-3 + cells for lungs** | 7 (5.25–8.75) | 36 (31.25–41) | 14.5 (13–17) | **0.000*** |
| **HPS for bowels** | 0 (0–0) | 3 (3–4) | 1 (1–1.75) | **0.000*** |

**MD (25–75):** Median (25–75 Interquartile range- IQR), **Casp-3⁺ cells**: Caspase-3 Positive Cell Count/unit **HPS**: Histopathological Scoring

*= Statistically significant.

## 4. Discussion

In lung injury, various studies have been conducted about the role of genetic markers such as *NF-κB, the cysteinyl aspartate-specific proteinase-8 (CASP8), Mitogen-activated protein kinase 1 (MAPK1), the cysteinyl aspartate-specific proteinase-3 (CASP3), Beclin-1 and Ribosomal Deoxyribonucleic Acid (rDNA)* genes in the maintaining cellular homeostasis and viability [7,43–46].

**Table 5. Binary comparison of the groups for both laboratory and clinical parameters.**

| | Groups | Control | NE | NE+EO |
|---|---|---|---|---|
| **BW** | **Control** | – | 0.980 | 0.960 |
| | **NE** | 0982 | – | 0.985 |
| | **NE+EO** | 0.961 | 0.985 | – |
| | | **Control** | **NE** | **NE+EO** |
| **LW** | **Control** | – | **0.000*** | **0.001*** |
| | **NE** | **0.000*** | – | **0.000*** |
| | **NE+EO** | **0.001*** | **0.000*** | – |
| | | **Control** | **NE** | **NE+EO** |
| **CIS** | **Control** | – | **0.000*** | **0.001*** |
| | **NE** | **0.000*** | – | **0.000** |
| | **NE+EO** | **0.001*** | **0.000*** | – |
| | | **Control** | **NE** | **NE+EO** |
| **BMS** | **Control** | – | **0.000*** | **0.000*** |
| | **NE** | **0.000*** | – | **0.000*** |
| | **NE+EO** | **0.000*** | **0.000*** | – |
| | | **Control** | **NE** | **NE+EO** |
| *TGFβ1* **Concentration Levels** | **Control** | – | **0.000*** | 0.758 |
| | **NE** | **0.000*** | – | **0.000*** |
| | **NE+EO** | 0.758 | **0.000*** | – |
| | | **Control** | **NE** | **NE+EO** |
| *MDA* **Concentration Levels** | **Control** | – | **0.000*** | 0.927 |
| | **NE** | **0.000*** | – | **0.000*** |
| | **NE+EO** | 0.927 | **0.000*** | – |
| | | **Control** | **NE** | **NE+EO** |
| *TGFβ1 E*xpresssion Levels | **Control** | – | **0.000*** | 0.266 |
| | **NE** | **0.000*** | – | **0.000*** |
| | **NE+EO** | 0.266 | **0.000*** | – |
| | | **Control** | **NE** | **NE+EO** |
| **HPS for lungs** | **Control** | – | **0.000*** | **0.000*** |
| | **NE** | **0.000*** | – | **0.000*** |
| | **NE+EO** | **0.000*** | **0.000*** | – |
| | | **Control** | **NE** | **NE+EO** |
| **Casp-3 + cells for lung** | **Control** | – | **0.000*** | **0.001*** |
| | **NE** | **0.000*** | – | **0.000*** |
| | **NE+EO** | **0.001*** | **0.000*** | – |

NE: Necrotizing Enterocolitis EO: Extra Virgin Olive Oil *TGFβ1:*Transforming growth factor beta-1 *MDA*: Malondialdehyde BW: Birth Weight LW: Last Day Weight HPS: Histopathological Scoring CIS: Clinical Illness Score BMS: Bowel Macroscopy Score Casp-3[+] cells: Caspase-3 Positive Cell Count/unit Area Statistical significant p-values are described as **dark color** punto.

*= Statistically significant.

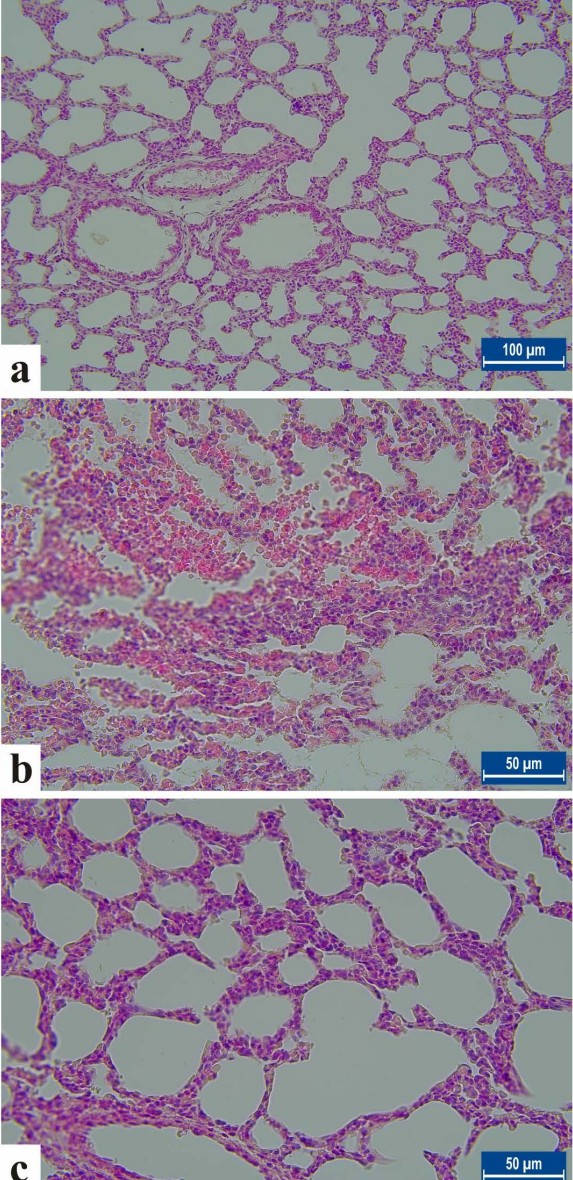

**Fig 2. Histological evaluation of the lung in neonatal rats from each experimental group.** A. Control group, Bar: 100μm, B. NE group showing increased neutrophil infiltration and severe lung tissue damage, C. Treatment group. Haemotoxylene-Eosin staining (H-E) Bar: 50μm.

To our knowledge, our study is the first that showed EO with rich phenol content administered by gavage in the hypoxia-hypothermia-hyperosmolar food-induced NE model of rats, which significantly reduces TGFß1 and MDA tissue homogenate levels, expression level of TGFß1 gene and tissue damage in the lung tissue ($p < 0.05$). We detected that in contrast to the LW, all of CIS, BMS and HPS were importantly higher in NE induction group than in healthy group. Because of one rat received 2 points, four rats received 3 points and three rats received 4 points in the HPS of the intestine, we concluded the adequacy of the model in creating NE.

It was reported that NE causes an increase in inflammation in the lungs, neutrophil infiltration, significant histopathological changes such as thickening and bleeding in the alveolar

septum [2,4,42,47–49], and an increase in apoptosis [2,47,48]. We detected significant histo-pathological lesions such as neutrophil infiltration, hyperemia, bleeding, and thickening of the interalvelor septum in the lung tissue of NE group. The HPS and Casp-3⁺ cells numbers in the lung tissue were significantly decreased in the NE + EO group compared to the NE group (p < 0.001, p < 0.001 respectively). We determined that enteral EO administration reduced apoptosis by decreasing Casp-3⁺ cells in the lung tissue and had a histological protective effect on NE-related lung damage.

Olive oil (olivea oleum), an important part of the Mediterranean diet, is grown throughout the Mediterranean basin and is obtained from olives [50] by mechanical pressing and contains relatively high amounts of polyphenolic [51]. The main components of EO are fatty acids (95–99%) composed of 55–83%monounsaturated fatty acids (MUFA) (especially oleic acid), 4–20% polyunsaturated fatty acids and 8–14% other fatty acids. The remaining part is minor polar compounds constitute 1–2% of EO and include tocopherols, phytosterols, squalene and phenolic compounds with important biological activities [52]. In EO, phenolic compounds are generally found in concentrations ranging from 50 to 940 mg/kg [53]. EO contains at least 36 different phenolics such as HTyr, Tyr, Ole, Ole aglycone, oleocanthral, flavonoids and lignans. The number and concentration of these compounds depend on the variety and maturity of the olive, the age of the tree, the region where it is grown, the time of harvest, the method of production and storage conditions. While refining EO by exposing to chemical processes does not cause any change in its fatty acid components, it causes a decrease or even disappearance

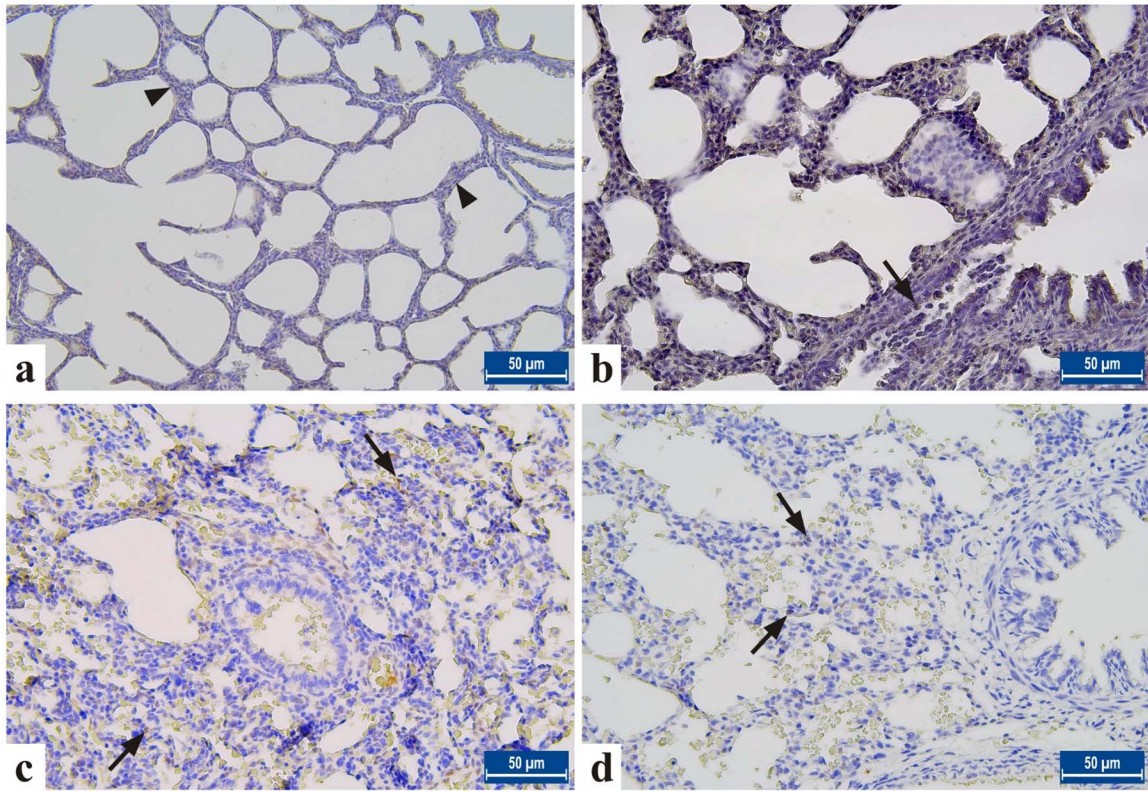

**Fig 3. Caspas-3 staining in lung tissue from neonatal rats in all groups.** A. Negative control. B. Control group. C. NE group, D. Treatment group. Arrowhead: Casp-3⁺ negative cells. Arrows: Casp-3⁺ cells. It is noteworthy that Casp-3⁺ cells were more numerous in the NE group than in the treatment group. Streptavidin-biotin-peroxidase method, Bar: 50μm.

**Table 6. Model Summary and Parameter Estimates for both concentration levels of *TGF1β*, MDA and clinical parameters.**

| Variable | Equation | Model Summary | | | | | Parameter Estimates | | | |
|---|---|---|---|---|---|---|---|---|---|---|
| | | R2 | F | df1 | df2 | sig | Constant | b1 | b2 | b3 |
| *TGFβ1 Conc Lev and* **BW** | Linear | ,059 | 1,388 | 1 | 22 | ,251 | 6,314 | -,001 | | |
| | Log | ,030 | ,688 | 1 | 22 | ,416 | 7,098 | -,185 | | |
| | Cubic | ,308 | 2,970 | 3 | 20 | ,056 | 5,011 | ,008 | −7,773 | −8,126 |
| *TGFβ1 Conc Lev and* **LW** | Linear | ,749 | 65,611 | 1 | 22 | ,000* | 9,432 | -,007 | | |
| | Log | ,736 | 61,302 | 1 | 22 | ,000* | 18,649 | −2,008 | | |
| | Cubic | ,749 | 19,935 | 3 | 20 | ,000* | 9,576 | -,008 | 1,243 | 3,695 |
| *TGFβ1 Conc Lev and* **CIS** | Linear | ,918 | 245,857 | 1 | 22 | ,000* | −3,629 | ,028 | | |
| | Log | ,930 | 293,366 | 1 | 22 | ,000* | −44,132 | 8,796 | | |
| | Cubic | ,938 | 100,268 | 3 | 20 | ,000* | −4,974 | ,031 | 3,551 | −6,781 |
| *TGFβ1 Conc Lev and* **BMS** | Linear | ,861 | 136,582 | 1 | 22 | ,000* | −1,456 | ,013 | | |
| | Log | ,891 | 179,204 | 1 | 22 | ,000* | −20,945 | 4,224 | | |
| | Cubic | ,896 | 57,348 | 3 | 20 | ,000* | −5,182 | ,045 | −6,460 | 3,461 |
| *TGFβ1 Conc Lev and MDA Conc Lev* | *Linear* | ,956 | 482,629 | 1 | 22 | ,000* | 52,715 | ,342 | | |
| | *Log* | ,911 | 223,937 | 1 | 22 | ,000* | −419,652 | 103,214 | | |
| | *Cubic* | ,961 | 164,019 | 3 | 20 | ,000* | 91,725 | ,001 | ,001 | −4,952 |
| *TGFβ1 Conc Lev and* **Casp−3 + cells** *for lung* | Linear | ,928 | 281,996 | 1 | 22 | ,000* | −3,787 | ,084 | | |
| | Log | ,934 | 312,198 | 1 | 22 | ,000* | −124,106 | 26,145 | | |
| | Cubic | ,938 | 100,491 | 3 | 20 | ,000* | −15,739 | ,184 | ,000 | 1,060 |
| *TGFβ1 Conc Lev and* **HPS** *for lung* | Linear | ,797 | 86,550 | 1 | 22 | ,000* | −1,198 | ,020 | | |
| | Log | ,823 | 102,367 | 1 | 22 | ,000* | −30,691 | 6,394 | | |
| | Cubic | ,824 | 31,295 | 3 | 20 | ,000* | −6,997 | ,070 | ,000 | 6,641 |
| *TGFβ1 Conc Lev and* **TGFβ1 Exp Lev** | Linear | ,840 | 115,681 | 1 | 22 | ,000* | 146,991 | 29,547 | | |
| | Log | ,605 | 33,684 | 1 | 22 | ,000* | 252,770 | 65,428 | | |
| | Cubic | ,868 | 43,684 | 3 | 20 | ,000* | 171,175 | −5,787 | 4,943 | -,165 |
| *MDA Conc Lev and* **BW** | Linear | ,058 | 1,356 | 1 | 22 | ,257 | 6,431 | -,002 | | |
| | Log | ,044 | 1,006 | 1 | 22 | ,327 | 7,752 | −,338 | | |
| | Cubic | ,177 | 2,259 | 2 | 21 | ,129 | 5,663 | ,000 | 6,624 | −2,621 |
| *MDA Conc Lev and* **LW** | Linear | ,764 | 71,196 | 1 | 22 | ,000* | 10,407 | −,019 | | |
| | Log | ,783 | 79,524 | 1 | 22 | ,000* | 23,204 | −3,156 | | |
| | Cubic | ,779 | 36,948 | 2 | 21 | ,000* | 12,148 | −,041 | 6,398 | ,000 |
| *MDA Conc Lev and* **CIS** | Linear | ,908 | 216,507 | 1 | 22 | ,000* | −7,650 | ,080 | | |
| | Log | ,934 | 311,018 | 1 | 22 | ,000* | −62,119 | 13,430 | | |
| | Cubic | ,940 | 165,323 | 2 | 21 | ,000* | −17,716 | ,210 | ,000 | ,000 |
| *MDA Conc Lev and* **BMS** | Linear | ,864 | 140,193 | 1 | 22 | ,000* | −3,409 | ,038 | | |
| | Log | ,893 | 183,230 | 1 | 22 | ,000* | −29,559 | 6,445 | | |
| | Cubic | ,901 | 95,706 | 2 | 21 | ,000* | −8,663 | ,106 | ,000 | ,000 |
| *MDA Conc Lev and* **Casp-3 + cells** | Linear | ,912 | 228,681 | 1 | 22 | ,000* | −15,677 | ,239 | | |
| | Log | ,926 | 276,983 | 1 | 22 | ,000* | −176,362 | 39,674 | | |
| | Cubic | ,926 | 132,102 | 2 | 21 | ,000* | −35,358 | ,493 | -,001 | ,000 |
| *MDA Conc Lev and* **HPS** *for lung* | Linear | ,796 | 86,037 | 1 | 22 | ,000* | −4,136 | ,058 | | |
| | Log | ,828 | 105,847 | 1 | 22 | ,000* | −43,810 | 9,771 | | |
| | Cubic | ,833 | 52,456 | 2 | 21 | ,000* | −12,416 | ,165 | ,000 | ,000 |

*(Continued)*

**Table 6.** (Continued)

| Variable | Equation | Model Summary | | | | | Parameter Estimates | | | |
|---|---|---|---|---|---|---|---|---|---|---|
| | | R2 | F | df1 | df2 | sig | Constant | b1 | b2 | b3 |
| *MDA Conc Lev and* **TGFβ**1 *Exp Lev* | Linear | ,805 | 90,958 | 1 | 22 | ,000* | −7,344 | ,080 | | |
| | Log | ,806 | 91,175 | 1 | 22 | ,000* | −60,393 | 13,120 | | |
| | Cubic | ,807 | 43,840 | 2 | 21 | ,000* | −9,645 | ,109 | −8,459 | ,000 |

*TGFβ1:*Transforming growth factor beta 1 *MDA*: Malondialdehyde **Conc**: Concentrations **Lev:**Levels **HPS:** Histopathological Scoring **CIS:** Clinical Illness Score **BMS:** Bowel Macroscopy Score **Casp-3⁺ cells:** Caspase-3 Positive Cell Count/unit Area **BW:** Birth Weight **LW**: Last Day Weight **\***= Statistically significant.

in the amount of phenolic compounds of it [51,52]. It has been shown that these phenolic compounds, which are responsible for the main biological properties of EO, have effects such as antioxidant [1,22,54,55], anti-inflammatory [1,22,55], antimicrobial [56], neuroprotective [57], anti-apoptotic [1,22] and antidysbiotic [58].

In the literature, many studies have been conducted on the therapeutic efficacy and relationship with molecular mechanisms of EO and its phenolic compounds. In rats with testicular damage induced by AlCl$_3$, while MDA levels increased, glutathione (GSH) and Catalase (CAT) activity decreased, but Tyr significantly improved these values in the testes. In addition, spermatological parameters such as motility, dead/live and abnormal spermatozoa ratios, testicular biopsy scores, both *CASP3* and *B-cell lymphoma gene-2 (Bcl-2)* expression levels were observed to improve significantly with Tyr treatment. It was detected that significantly decreased expression levels of *Nuclear factor erythroid 2-related factor 2 (Nrf-2),* which induces the production of many antioxidants and detoxification enzymes, and *Heme oxygenase-1 (HO-1)*, a protective factor against OS, caused by AlCl$_3$ application, reached the levels measured in the control group with Tyr treatment. As a result, oral Tyr treatment for 10 weeks in AlCl$_3$-induced testicular damage alleviates oxidant stress, increases antioxidant activity and reduces apoptosis by inducing the Nrf-2/HO-1 signaling pathway [28].

In a study investigating the effect of phenolics secoiridoids such as HTyr, Ole aglycone in EO on *TGFB1*-induced fibrogenic and oncogenic EMT in Madin–Darby canine kidney (MDCK) cells and human breast cancer (MCF-7) cells, it was determined that the expression of the epithelial marker E-cadherin decreased and the expression of the mesenchymal marker Vimentin increased with *TGFß1* induction in MDCK cells, and then after the cells were treated with EO, the expression of E-cadherin increased significantly and the expression of Vimentin was significantly inhibited. Also *Snail family transcriptional repressor 2 (SNAI2)*, a transcription factor that promotes EMT, is up-regulated by *TGFß1* induction in MDCK cells, but EO treatment reverses this effect. The cellular changes observed in *TGFβ1*-induced MCF-7 breast cancer epithelial cells resembled tumor cells undergoing EMT, E-cadherin expression, which was decreased in *TGFβ1*-induced cells, was increased by EO treatment. Also, *SMAD4* and *SNAI2* expression, which promote EMT, were upregulated by *TGFß1*, while *SMAD4* and *SNAI* expressions were suppressed by EO treatment. In conclusion, it was reported that phenolics secoiridoids such as HTyr, Ole aglycone in EO, act as anti-aging phytochemicals by inhibiting *TGFB1*-induced fibrogenic and oncogenic EMT in MDCK cells and MCF-7 cells [33].

Olive leaf extract containing 25% Htyr given orally for 1 week after induction of colitis in albino rats with rectal acetic acid reduced mortality rate and disease activity indices. While a significant decrease was observed in nitrite oxide (NO), MDA and myeloperoxidase (MPO), conversely a significant increase was detected in superoxide dismutase (SOD), CAT and glutathione peroxidase (GSH-Px) activities in the colon tissue of the treatment group.

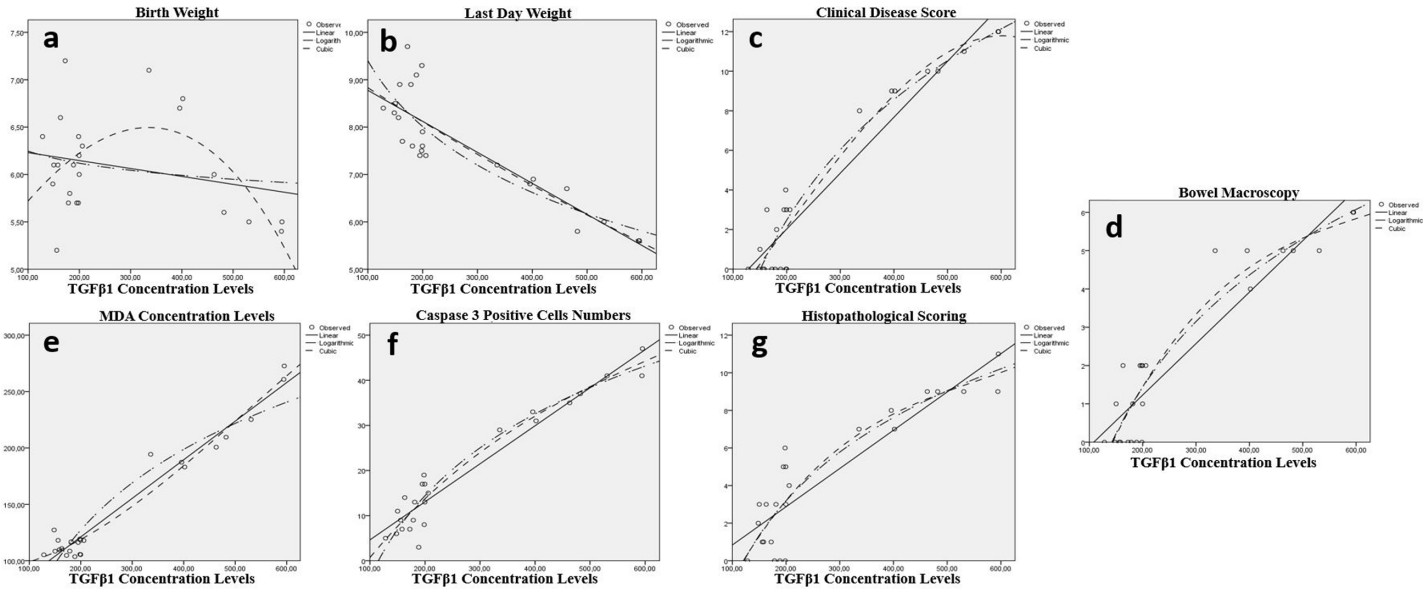

**Fig 4. Relation between the TGFβ1 concentration levels in lung tissue and clinical parameters of the groups.** There were statistically significant relation between TGFβ1 concentration levels and all of last day weight (b), clinical disease score(c), bowel macroscopy(d), MDA concentration levels (e), Caspase-3 positive cell count/ unit (f) and histopathological scoring (H&E scored 0–12) (g). Conversely the relation between TGFβ1 concentration and birth day was not significant (a).

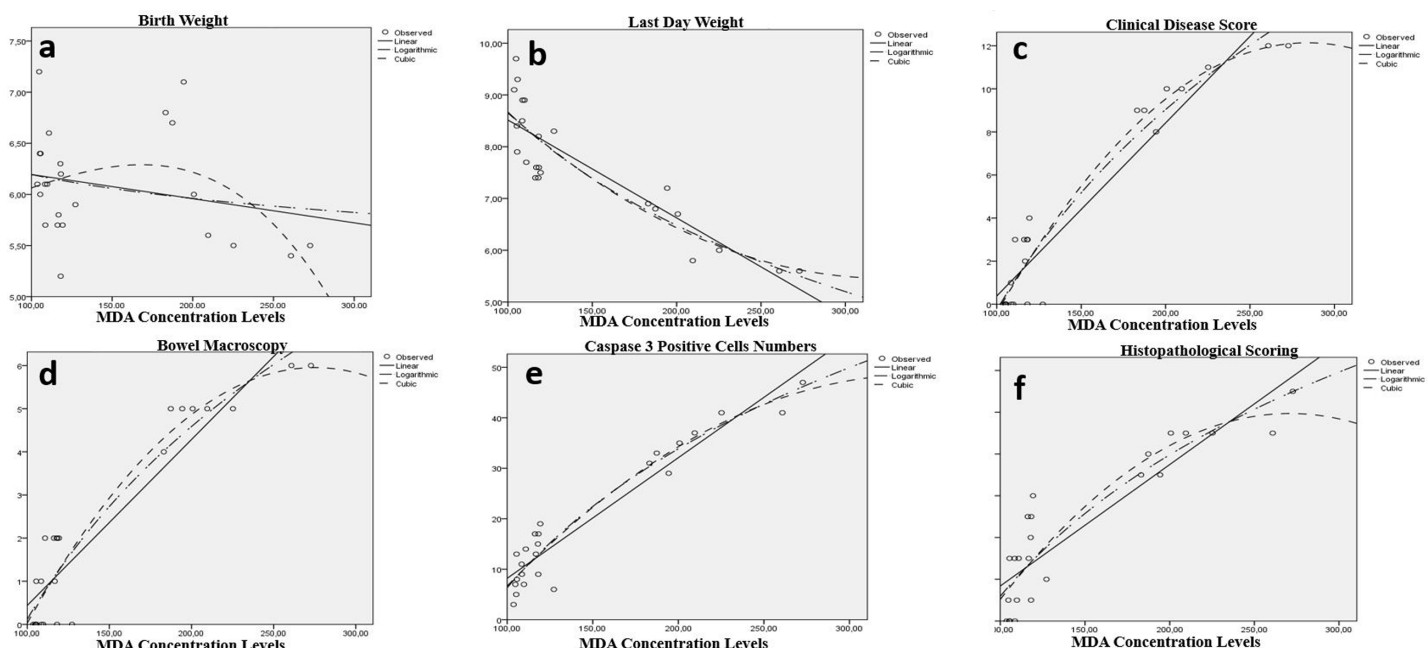

**Fig 5. Relation between the MDA concentration levels in lung tissue and clinical parameters of the groups.** There were statistically significant relation between MDA concentration levels and all of last day weight (b), clinical disease scor(c), bowel macroscopy(d), Caspase-3 positive cell count/unit (e) and histopathological scoring (H&E scored 0–12) (f). Conversely the relation between MDA concentration and birth day was not significant (a).

Additionally, significant decreases were detected in the expression levels of proinflammatory cytokines (PIC) (*IL-1β, Tumor necrosis factor-alpha (TNF α), IL10*), *cyclooxygenase-2 (Cox-2), Inducible nitrite oxide synthase (iNOS), TGFβ1, monocyte chemoattractant protein-1 (MCP1)* and *nuclear factor kappa-B (NF-κB)*. Also, the apoptotic gene *Bcl-2-associated X protein* (*Bax*) was downregulated, while the anti-apoptotic gene *Bcl-2* was upregulated. As a result, it was reported that olive leaf extract enriched with HTyr reduced OS, inflammation and apoptosis in colitis-induced rat colon tissue [22]. In another colitis model induced with oral dextran sulfate, EO given by gavage for 11 days alleviated the clinical findings in mice and significantly decreased the gene expression levels of *IL-1β*, *TGFβ1* and *IL-6* in colon tissue. In the histopathological examination, EO improved intestinal permeability and histopathological damage caused by inflammation [23].

In a liver fibrosis model induced by oral CCl4 administration for 8 weeks of rats, the CCl4 administration increased OS in the rat liver and caused significant increases in MDA levels, whereas EO administration during the induction period caused significant decreases in MDA levels by reducing lipid peroxidation. The *TGFβ1, Toll Like Receptor 4 (TLR4), Reduced nicotinamide adenine dinucleotide phosphate (NADPH) oxidase* and *NF-κB* expression levels were increased in the CCl4 group compared to the control group, but after the EO treatment these

**Table 7. Model Summary and Parameter Estimates for expression levels of *TGFβ1* with all of Caspase 3, MDA and clinical parameters.**

| Variable | *Equation* | Model Summery | | | | | Parameter Estimates | | | |
|---|---|---|---|---|---|---|---|---|---|---|
| | | *R2* | *F* | *df1* | *df2* | *sig* | Constant | *b1* | *b2* | *b3* |
| *TGFβ1 Exp Lev and **BW*** | Linear | ,077 | 1,836 | 1 | 22 | ,189 | 6,217 | –,031 | | |
| | Log | ,007 | ,151 | 1 | 22 | ,702 | 6,092 | –,024 | | |
| | Cubic | ,272 | 2,491 | 3 | 20 | ,090 | 5,991 | ,307 | –,048 | ,002 |
| *TGFβ1 Exp Lev and **LW*** | Linear | ,744 | 63,802 | 1 | 22 | ,000* | 8,543 | –,210 | | |
| | Log | ,595 | 32,346 | 1 | 22 | ,000* | 7,799 | –,491 | | |
| | Cubic | ,747 | 19,663 | 3 | 20 | ,000* | 8,473 | –,091 | –,020 | ,001 |
| *TGFβ1 Exp Lev and **CIS*** | Linear | ,862 | 137,513 | 1 | 22 | ,000* | ,316 | ,883 | | |
| | Log | ,717 | 55,620 | 1 | 22 | ,000* | 3,425 | 2,100 | | |
| | Cubic | ,864 | 42,442 | 3 | 20 | ,000* | ,390 | ,647 | ,053 | -,003 |
| *TGFβ1 Exp Lev and **BMS*** | Linear | ,807 | 92,098 | 1 | 22 | ,000* | ,421 | ,419 | | |
| | Log | ,701 | 51,698 | 1 | 22 | ,000* | 1,890 | 1,020 | | |
| | Cubic | ,811 | 28,594 | 3 | 20 | ,000* | ,299 | ,588 | –,022 | ,001 |
| *TGFβ1 Exp Lev and **MDA** Conc Lev* | Linear | ,805 | 90,958 | 1 | 22 | ,000* | 102,955 | 10,118 | | |
| | Log | ,560 | 27,959 | 1 | 22 | ,000* | 139,318 | 22,014 | | |
| | Cubic | ,844 | 35,956 | 3 | 20 | ,000* | 110,388 | 1,632 | ,811 | -,008 |
| *TGFβ1 Exp Lev and **TGFβ1** Conc Lev* | Linear | ,840 | 115,681 | 1 | 22 | ,000* | 146,991 | 29,547 | | |
| | Log | ,605 | 33,684 | 1 | 22 | ,000* | 252,770 | 65,428 | | |
| | Cubic | ,868 | 43,684 | 3 | 20 | ,000* | 171,175 | –5,787 | 4,943 | -,165 |
| *TGFβ1 Exp Lev and **Caspase-3 + Cells*** | Linear | ,805 | 90,958 | 1 | 22 | ,000* | 102,955 | 10,118 | | |
| | Log | ,560 | 27,959 | 1 | 22 | ,000* | 139,318 | 22,014 | | |
| | Cubic | ,844 | 35,956 | 3 | 20 | ,000* | 110,388 | 1,632 | ,811 | -,008 |
| *TGFβ1 Exp Lev and **HPS** for lung* | Linear | ,857 | 131,707 | 1 | 22 | ,000* | 1,450 | ,680 | | |
| | Log | ,723 | 57,351 | 1 | 22 | ,000* | 3,841 | 1,630 | | |
| | Cubic | ,860 | 40,859 | 3 | 20 | ,000* | 1,313 | ,842 | –,016 | ,000 |

*TGFβ1:*Transforming growth factor beta 1 **MDA**: Malondialdehyde **Conc**: Concentrations **Lev:**Levels **BW**: Birth Weight **LW**: Last Day Weight **HPS**: Histopathological Scoring **BMS**: Bowel Macroscopy ScoreF **CIS**: Clinical Illness Score **Casp-3⁺ cells:** Caspase-3 Positive Cell Count/unit Area**\*** = Statistically significant.

increased mRNA expressions were significantly decreased. In the western blot analysis, the levels of NADPH oxidase, NF-κB and TGFβ1 proteins, which were increased by CCl4 induction in the liver, were significantly reduced by EO treatment and as a result, it was reported that EO suppressed OS and inflammation [27]. In rats with hepatic toxicity induced by oral fluoxetine, both EO and olive leaf extract administered orally showed anti-inflammatory effects by significantly reducing Alanine Transaminase (ALT), Aspartate Transaminase (AST), alkaline phosphatase (ALP) and PIC (*TNF-α* and *IL-1β*) that increased in the circulation with fluoxetine induction. Histopathological analysis showed that abnormal histological changes accompanying inflammatory cell infiltration in the liver tissue were improved by the application of EO and olive leaf extract. EO and olive leaf extract reduce lipid peroxidation by causing significant decreases in MDA and NO levels in liver tissue, and also show antioxidant effects by significantly increasing the activities of SOD, CAT and GSH-Px, which are antioxidant enzymes that have decreased due to induction in liver tissue. The gene expression levels of *Bax* and *CASP3*, which increased with fluoxetine induction, decreased significantly, while the gene expression level of decreased *Bcl-2* increased significantly with these treatments. As a result, it has been reported that both EO and olive leaf extract exhibit anti-inflammatory properties, reduce OS, increase antioxidant effect and also prevent apoptosis [59].

It was shown that triterpenic acids rich EO attenuated systolic and diastolic blood pressure increases in spontaneously hypertensive rats, increased *endothelial nitric oxide synthase* (*eNOS*) expression in aortic tissue, reduced *TNF-α* levels, and collagen accumulation in the arterial wall by causing a significant decrease in *TGFB1*, which regulates collagen accumulation [30].

Inflammation and OS has crucial role in the pathogenesis of lung injury, and infections, sepsis, environmental pollutants, smoking, allergens and genetic predisposition cause lung damage [60]. Since EO and phenolic compounds have the above-mentioned properties, they

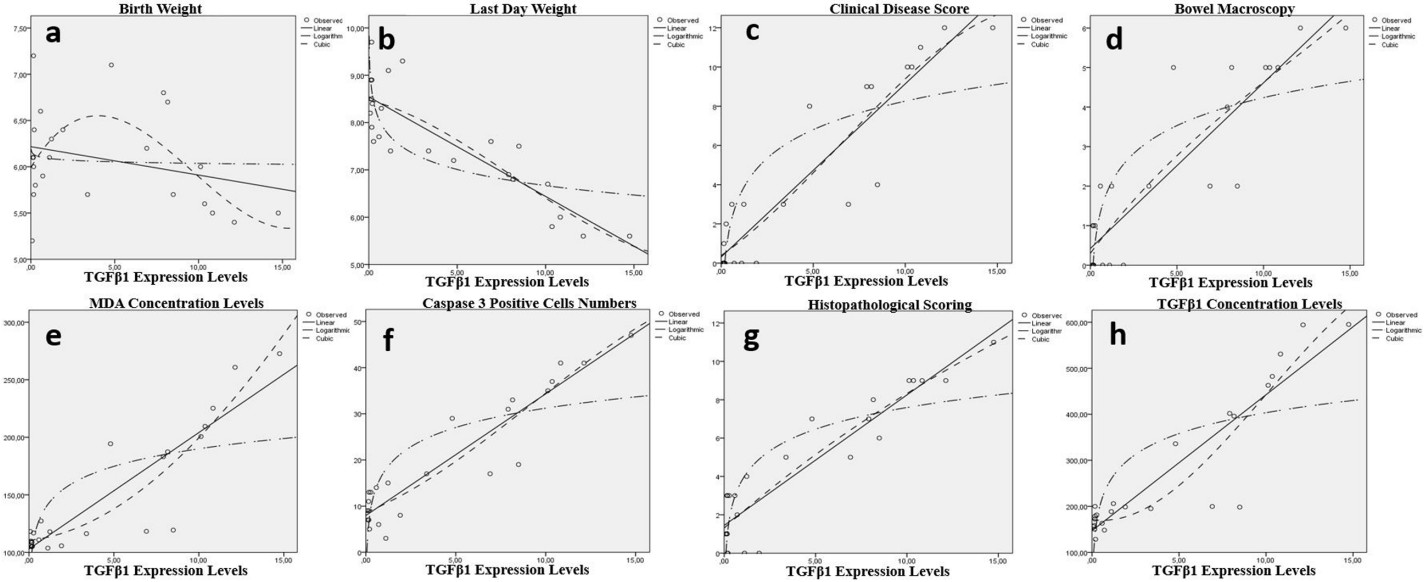

**Fig 6. Relation between the *TGFβ1* expression levels in lung tissue and clinical parameters of the groups.** There were statistically significant relation between *TGFβ1* expression levels and all of last day weight (b), clinical disease scor(c), bowel macroscopy(d), MDA concentration levels (e), Caspase-3 positive cell count/unit (f), histopathological scoring (H&E scored 0–12) (g) and *TGFβ1* concentration levels (h). Conversly the relation between *TGFβ1* expression levels in lung tissue and birth day was not significant (a).

prevent lung damage by suppressing excessive amounts of PIC and OS in the pathogenesis of lung damage.

IL-17A induces OS, DNA damage and apoptosis in the A549 human lung epithelial cell line in vitro, but Ole reverses these effects [61]. In a study examining the both effects of enterally administered Ole and silver nanoparticle-loaded Ole on lung tissue for 11 days on intraperitoneal LPS-induced ALI in albino rats, a significant decrease MDA and increase GSH and SOD activities were observed in both treatment groups. There was an important decrease the levels of PIC (TNF-α, IL-6 and IL-1β) in the rats given Ole treatment. These PIC were higher in the silver-loaded Ole treatment than the only Ole treated group, but were significantly reduced compared to the LPS induction rats. In addition, *P2X purinoceptor 7 (P2X7R), Bax, IL-1β*, *TNF-α, and TLR4* gene expressions were importantly higher in LPS-induced rats, and the expressions of these genes were significantly decreased in both treatment groups. The histological appearances in the lung were similar, and that both Ole and silver-loaded Ole treatment reduced OS, inflammation, tissue damage and apoptosis in the control and both treatment groups [62].

In ALI, which occurred after intratracheal LPS administration to rats for 20 days, it was seen that while neutrophils in the BALF and MDA levels increased in serum and lung tissue, the GSH levels, GSH-Px and CAT activities decreased. A significant increase in IL-6, TNF-α, MPO and NF-kB was observed in BALF, serum and lung tissue, in addition to interalveolar septum widened due to capillary hyperemia, edema and Polymorphonuclear leukocyte (PMNL) infiltration in the lungs after LPS induction. The vascular-bronchial interstitial edema, congestion and intense inflammatory cell accumulation occured. Oral administration of Ole simultaneously with LPS induction positively affects the above-mentioned biochemical and pathological changes in serum, BALF and lung tissue, and as a result, Ole causes significant improvements in the histological damage of the lung by reducing OS and inflammation [17].

In a study examining the effect of HTyr on lung damage caused by nasal LPS induction, it was found that HTyr reduced lung edema, inflammatory cells in BALF and lung tissue, and strongly regulated PIC as well as LPS-stimulated sirtuin (SIRT) expression and autophagy inhibition, Mitogen-activated protein kinase *(*MAPK) phosphorylation regulated by HTyr and ameliorates lung injury [18]. Exposure to 85dB noise and toluene inhalation for 6 weeks caused lung damage in rats, increased inflammation and OS, and olive extract given orally once a day for 6 weeks reduced MDA levels. It increases SOD and CAT activities and prevents lung damage by suppressing OS and inflammation by reducing PIC such as TNF-α and IL-1β [63]. In the study evaluating the effect of Tyr on ALI caused by LPS induction in mice, it was found that Tyr reduced PIC increased in BALF and lung tissue as a result of LPS induction. The activations of inflammatory molecules such as increased COX-2, iNOS and phosphorylated-IκBα were suppressed by the presence of Tyr in lung tissue. The increase in MPO and the decrease in SOD activities caused by LPS were inhibited by Tyr. In addition, Tyr reduced the expressions of *iNOS, COX-2* and PIC and the nuclear translation of *NF-kB* in LPS-stimulated RAW 264.7 macrophage, resulting in Tyr ameliorating LPS-related ALI, inhibiting NF-κB, reducing the release of PIC and it may be a potential therapeutic agent in inflammatory lung diseases [16]. In another study investigating the effect of Try on another LPS-induced mouse lung injury model, Tyr improved pulmonary permeability and histopathological changes, decreased the expression of PIC and increased the expression of antioxidant enzymes [19]. Pulmonary fibrosis is a response to endothelial and epithelial cells that are frequently damaged by the invasion of inflammatory cells. In the rat study investigating the effects of Ole on pulmonary fibrosis induced by intratracheal bleomycin, TNF-α, IL-13, TGFβ1, Platelet-derived growth factor (PDFG), lung collagen index and MDA increased significantly in BALF, but Ole prevented pulmonary fibrosis by reversing these effects [21].

When the above detail studies conducted to show the effect of EO on markers known to be effective in anti-inflammatory (IL-1β, TNF α, IL10, IL-6, IL-17A, TGFB1, MAPK, Cox-2, iNOS, MCP1, NF-κB, eNOS, TLR4, NADPH, NF-κB, P2X7R, SIRT, IL-13, Nrf etc.), antioxidant (MDA, GS, CAT, Nrf-2, HO-1, NO, MPO, SOD, CAT, GSH-Px etc.) and antiapoptotic (CASP3, Bcl-2, Bax etc.) pathways to be taken into consideration, these clearly indicate the anti-inflammatory and antioxidant capacity of EO.

NE is one of the leading causes of gastrointestinal-related mortality in premature babies [64]. Incomplete development of the intestinal mucosal barrier and immune system due to prematurity, increased OS as a result of inadequate antioxidant capacity, dysbiosis, and feeding with osmolar formulas are among the most important predisposing risk factors for NE [65]. While the severity of NE in the intestine is an important determinant of initial mortality in preterms with NE, long-term complications due to NE related the severity of other organ involvement, especially the lungs [66]. In particular, NE-related lung damage is more severe and more difficult to treat than BPD, which is frequently observed in the premature neonatal period [2,4]. The pathogenesis of NE-induced lung damage remains unknown, but current ongoing research focuses on explaining and preventing the mechanisms that lead to NE-related lung damage, which is an important cause of mortality and complication, in addition to studies on the etiopathogenesis and prevention of NE.

*TLR4*, which plays an important role in the innate immune response to pathogens, is found in the intestine and lung, and *TLR4* activation plays a role in the intestinal and lung damage that occurs in NE [4]. *NF-κB* is activated by the *TLR4* signaling transduction pathway, passes into the nucleus and causes increased gene expression of PIC [67]. In mice with NE induction, significant histopathological damage, increased neutrophil infiltration and MPO expression occured, but these effects were significantly improved with orally administered bovine milk exosome treatment. *TLR4* and *NF-κB* expression significantly increased in the lungs of the NE group, conversely to the exosome treated group. While the level *of Nod-like receptor protein 3* (*NLRP3*) mRNA expression increased in the lungs of NE group, exosome treatment did not reduce this increase, but the increased NLRP3 protein level in the NE group was significantly reduced with exosome treatment. Also significantly increased caspase 1 protein expression, another component of the NLRP3 inflammasome, was significantly decreased in the exosome-treated group. Results suggest that milk-derived exosomes have the ability to regulate NE-associated lung injury by inhibiting the *NF-κB* inflammatory pathway and *NLRP3* inflammasome activity [48].

In a study using human bronchial epithelial cell line (HBE 135-E6E7), mice and human tissues to elucidate the pathogenesis of NE-associated lung injury and to evaluate the efficacy of aerosolized C34, a *TLR4* receptor inhibitor, it was shown that *TLR4* expression in lung tissue gradually increases after birth and that mice and premature infants with NE-associated lung injury express higher levels of pulmonary *TLR4* than age-matched controls. The expression of proinflammatory molecules *iNOS*, *IL-8*, *TLR4* and histopathological damage are significantly increased in both the intestines and lungs of premature infants with NE and NE-induced mice. In mice with transgenic *TLR4* deletion in the intestinal epithelium, no damage was observed in the intestines and lungs as a result of NE induction, whereas in mice with transgenic *TLR4* deletion in the lung epithelium, damage developed in the intestines but not in the lungs when NE was induced. As a result, it was reported that pulmonary *TLR4* has a role in NE-associated lung injury and that *TLR4* signaling is required in the intestines and lungs. *TLR4* activation in the intestinal epithelium of wild-type mice with NE induced the release of High mobility group box 1 (*HMGB1*) protein, a proinflammatory molecule, but did not induce *HMGB1* release in mice with TLR4 transgenically deleted in the intestinal epithelium, and the release of *HMGB1* into the circulation in NE is dependent on *TLR4* signaling in the intestinal epithelium. *HMGB1*

released into the circulation activates pulmonary epithelial *TLR4*, inducing *CXC-Chemokine Ligand-5* (*CXCL5*), a chemotactic cytokine for neutrophils, and thus causing lung injury due to the accumulation of proinflammatory neutrophils in the lung tissue. Also aerosolized application of the *TLR4* inhibitor C34 failed to prevent damage in the intestines of NE-induced mice, but significantly improved the histological appearance in the lung tissue and significantly reduced the expression of *IL-6*, *CXCL1* and *CXCL5* in the lung tissue and the percentage of neutrophils in BALF. Finally, it has been reported that C34 interrupts the *TLR4* mediated neutrophil recruitment cascade in lung tissue [4].

In a study on the effects of Extracellular cold-inducible RNA-binding protein (eCIRP) on the severity of NE-associated intestinal injury and the effects of treatment with eCIRP scavenger peptide milk fat globule-epidermal growth factor VIII (MFG-E8) derived oligopeptide 3 (MOP3) [68], eCIRP levels in prospective stool samples of neonates with NE were reported to be 3.1-fold higher than in age-matched healthy controls. After NE induction, in addition to 64-fold increased systemic eCIRP levels in wild-type mice compared to mice with transgenic deletion of CIRP, *IL-6, TNF-α, IL-1β* gene expression and histopathological damage increased in lung tissue of wild-type mice, NE-compatible damage occurred in their intestines whereas no increase was observed in transgenic mice. It has been reported that apoptotic cells are significantly reduced in transgenic mice compared to wild-type mice and that eCIRP contributes to lung inflammation and injury in NE. In addition, MOP3 treatment reduces the severity of NE in the intestines by regulating systemic eCIRP, which increases as a result of NE induction. In lung tissue, MOP3 treatment has been reported to be effective in NE-associated lung injury because it significantly reduces histopathological damage and the number of apoptotic cells, as well as decreasing the mRNA levels of *IL-6, TNF-α* and *IL-1β* [69].

It was observed that the mRNA expression levels of *IL-6*, *IL-1β* and *TNF-α* increased in both intestine and lung tissue of NE-induced mice and caused histopathological damage. The significant increase in the number of neutrophils in the alveolar interstitium of NE-associated lung tissue and in neutrophil elastase and MPO secreted by neutrophils suggest that neutrophils are important in the pathogenesis of lung injury. Stimulation of neutrophils purified from lung tissues with N-formyl-methionyl-leucy1-phenylalanine (fMLP), a neutrophil activator, showed a significant increase in reactive oxygen species (ROS) in the NE group compared to the control group. The N-Asetil Sistein (NAC) treatment administered to mice via the nasal cavity during the NE induction period resulted in significant decreases in histopathological damage, neutrophils, neutrophil elastas and MPO as well as mRNA expression levels of *IL-6*, *IL-1β* and *TNF-α* in intestinal and lung tissues. Kelch-like ECH-associated protein 1 (Keap1) and *Nrf2* gene expression levels decreased in NE-associated lung-injured tissue, and significant increases in these decreased gene expressions were recorded with NAC treatment. The mRNA expression levels of *HO-1, SOD1* and *NAD(P)H quinone oxidoreductase 1 (NQO1)* genes controlled by the Keap1-Nrf2 pathway were decreased in NE-related lung tissue, but these levels are increased with NAC treatment. So it was concluded that the Keap1-Nrf2 pathway ultimately contributes to NEC-induced lung damage [70].

In another NE model, it was shown that lung and intestinal damage were more severe and IL-6 levels were higher in eNOS knockout mice than in wild-type mice. This effect was due to eNOS increasing NO synthesis in the mesentery, which indicates that it also modulates IL-6 release in addition to its antioxidant and vasodilator effects [42]

Umbilical mesenchymal stem cells have been shown to reduce disease presumably via paracrine release of $H_2S$ in animal models of NE [71]. $H_2S$, a gasotransmitter secreted from umbilical mesenchymal stem cells during oxidative stress, is synthesized in cells by cystathionine-β-synthase, cystathionine-γ-lyase and 3-mercaptopyruvate sulfurtransferase (3-MPST) enzymes [72]. In a study using NE-induced mouse and human umbilical mesenchymal stem cells,

inhibition of these enzymes was achieved by siRNA transfection in cultured human umbilical mesenchymal stem cells. It has been reported that in vitro, there is more $H_2S$ production in normal stem cells under hypoxia, but $H_2S$ production is reduced in cells with enzyme inhibition. It was also reported that in mice treated intraperitoneally with negative control siRNA stem cells, clinical disease scores as well as intestinal and lung histological damage scores were significantly improved compared to the NE group, but in the group treated with enzyme inhibited stem cells, these parameters were significantly worse compared to the negative control siRNA group. As a result, it has been reported that $H_2S$ is protective in cases where oxidative stress occurs in relation to the mentioned pathways [73].

Hydrogen Sulfide ($H_2S$) is an important agent that causes a vasodilator effect by causing an increase in NO in tissues via eNOS and has important properties such as antioxidant, angiogenetic, and cell proliferation [74]. Both lung and intestinal damage was observed in both wild-type mice and eNOS$^{C440G}$ mutant mice with genetic ablated *eNOS* gene after NE induction, and that intraperitoneal administration of GYY4137, $H_2S$ donor, during the induction period reduced lung and intestinal damage in wild-type mice, while this effect was not observed in eNOS$^{C440G}$ mutant mice. $H_2S$ has a NO-dependent vasodilator effect, reduces ROS production, may have protective effects on the lung and intestine by protecting the development of alveolar and lung vascular networks, and as a result, $H_2S$ provides its protective effects in experimental necrotizing enterocolitis by using Cysteine 440 on eNOS [75]. In a study investigating NE-related lung damage in mice and the effect of $H_2S$-Mesalamine, H2S donor, on lung damage, NE induction significantly increase TLR4 and IL-6 levels in the intestinal and lung tissues of both wild-type and transgenically *eNOS* gene knockout mice. It was observed that $H_2S$ derivative treatment attenuated the increase of TLR4 and IL-6 in wild-type mice and induced clinical and histological improvement, but did not have these effects in transgenic mice. As a result, it has been reported that eNOS plays an important role in NE-related lung damage and that H2S-Mesalamine acts through eNOS [49].

In a study evaluating NE-related lung injury in mice and the effect of formula enriched with digested fat, NE induction resulted in increased *TNF-α* and *lipocalin-2* (*Lcn-2*) gene expressions, and 3'-nitrotyrosine (3'-NT), an indicator of OS, and the transcription of the superoxide-producing NADPH oxidase 2 ($NOX_2$) enzyme in lung tissue. Histologically, alveolar destruction, vascular extravasation and PMNL accumulation occur in the lung, and immunohistologically, MPO activity, the expression of neutrophil-specific enzyme ELANE and *pro-apoptosis gene p53 up-regulated modulator of apoptosis* (*PUMA*) gene are increased and intense apoptosis has been reported to be observed. Also, the building block proteins SP-A and SP-D of surfactant secreted by epithelial type 2 cells, which are an important structure in the defense of the lung, were significantly reduced by immunostaining method. In addition, Zonula Occludens (ZO)-1, a tight junction protein in lung tissue, causes a significant decrease in immunostaining and quantity [47].

In mice and humans with NE, a decrease in T regulatory lymphocytes (Treg), which are involved in the accumulation of Th17 proinflammatory leukocytes and the maintenance of immune homeostasis, has been reported to be associated with severe lung injury. It has also been reported that administration of CD4 + T lymphocytes isolated from NE-induced mouse lungs to the lungs of immunosuppressed mice causes severe lung injury, that depletion of Tregs exacerbates NE-associated lung injury, and that consequently, Th17/Treg imbalance is required for the induction of NE-associated lung injury. Furthermore, selective deletion of TLR4 from surfactant protein C-1 (SFTPC1) pulmonary epithelial cells has been reported to reverse lung injury, while TLR4 activation has been reported to induce chemokine (C-C) ligand 25 (CCL25), which mediates chemotaxis of Th17 in mouse lungs as a result of NE induction. Aerosolized inhibition of CCL25 and TLR4 has also been reported to restore Tregs

that attenuate NE-associated lung injury. As a result, it was reported that TLR4 activation in SFTPC1 cells disrupts the Treg/Th17 balance in the lungs via CCL25 and causes NE-associated lung injury, while TLR4 inhibition protects against NE-associated lung injury by stabilizing the Th17/Treg balance in the lungs [76].

In summary, it is predicted that preventing the inflammatory response, OS and chemotaxis of inflammatory cells to the lungs may prevent NE-related lung damage. OS, inflammation, structural and functional factors, and apoptosis appear to be involved in the pathogenesis of NE-associated lung injury.

In our study important differences were detected among the groups for MDA lung tissue concentration levels ($p < 0.05$). Immunohistochemically, in all groups, CASP3 activity was evaluated to detect the apoptosis process in the lung and *CASP3* expression was observed to be increased in the NE group and significantly decreased in the NE+EO group ($p < 0.001$) (Table 4, 5 Fig 3B, 3C, 3D). Also, statistically meaningful relation between MDA levels and all of LW, CIS, BMS, Casp-3$^+$ cells and HPS for lung were detected.

*TGFβ1* is a pleiotropic cytokine that plays a role in many physiological and pathological processes such as cellular differentiation, migration, apoptosis, regulation of immune systems, immune tolerance, hemostasis, fibrosis, inflammation, cancer progression, and its effects depend on the tissue and agent [77]. Its opposing effects, both suppressing and activating inflammation, are important in maintaining immunological balance [78]. *TGFβ1* promotes healing by inhibiting the immune response and uncontrolled inflammation. However, *TGFβ1* serves as one of the main mediators and initiates fibrosis and apoptosis endothelial cells that can cause tissue damage if left uncontrolled, and in infective cases the role of *TGFβ1* is largely tissue and pathogen dependent [79]. Although the pathogenesis of *TGFβ1*-mediated lung damage is not fully known, studies have reported that *TGFβ1* causes damage to the lungs by increasing lung microvascular alveolar permeability and increasing actin stress fiber formation [80]. In different studies, various pathways that affect *TGFβ1* and/or are affected by *TGFβ1* have been stated. In ALI induced mice after nickel inhalation, while a decrease in the expression of genes that play important roles in alveolar fluid resorption, synthesis or reuse of surfactant proteins and phospholipids, a significant increase of *TGFβ1* levels and gene expression levels in BALF were found. As a result, *TGFβ1* is a central mediator of ALI [81]. While miR-1258 expression and SOD activities were decreased, *TGFβ1* expression, p-SMAD3 protein levels, histological changes indicating ALI, IL-6, IL-1β, TNF-α, MDA level, *PBX/knotted 1 homeobox 1* (*Pknox1)* gene expression and protein levels were increased in septic ALI patients, in LPS-induced Immortalized human bronchial epithelial cell line *(*BEAS)-2B cells and LPS induced mice, *miR-1258* expression treatment inhibit inflammation and OS in ALI through the *TGFβ1*/SMAD3 cascade regulated by *Pknox1* [10]. It has been reported that *TGFβ1* expression increases in fibrosis of human lung and bleomycin-induced mouse lung tissues. *TGFβ1* increased the expression of ornithine aminotransferase (OAT) in fibroblasts, and the increase in OAT expression regulated both *TGFβ1*-induced and constitutive expression of collagen, fibronectin and alpha-smooth muscle actin (α-SMA), which are important components of fibrosis, so OAT is important in lung fibrogenesis. Also OAT causes OS as a result of the production of ROS in the mitochondria by activating proline dehydrogenase, and OAT inhibition with L-canaline suppresses *TGFβ1* activity and signaling pathway. As a result, it was observed that OAT could regulate *TGFβ1* expression and that OAT inhibition suppressed the Smad-dependent and Smad-independent signaling pathways induced by *TGFβ1* [13]. Severe Coronavirus disease 2019 (COVID-19) is characterized by excessive release of PIC leading to acute respiratory distress syndrome (ARDS) and potentially life-threatening systemic inflammation. In ARDS patients, *TGFβ1* increases in the parenchyma and blood in the early stages, and in the later stage, chronic inflammation dominated by *TGFβ* signaling and Immunglobulin-A2

production develops, which induces fibrosis [82]. In a study a significant increase in *TGFβ1*, C-Reactive Protein (CRP) and D-Dimer levels, and a positive correlation between *TGFβ1* and CRP were detected in Covid-19 patients compared to the control group. As a result, it has been reported that *TGFβ1* levels are the main molecule for the pathophysiology of the disease and are a marker that can be used in the early diagnosis of COVID-19 and the course of the disease [83]. *TGFß1*, which is a predisposing risk factor for the severity of COVID-19, has a profibrotic effect [84] and is associated with disease severity. In a study, serum *TGFβ1* levels showed a positive correlation with inflammatory markers such as CRP, leukocyte count, absolute neutrophil count, platelet count, fibrinogen in all patients. The higher serum *TGFβ1* concentrations, which increased with the severity of COVID-19, were measured and *TGFβ1* levels at hospitalization is distinctive in predicting the severity and development of complications.

In the study investigating the effect of Tyr, one of the phenolics of EO, on the nonalcoholic steatohepatitis model, steatosis and hepatic fibrosis and increased *TGFβ1* gene expression were observed. Tyr reversed these effects and had a positive effect on nonalcoholic steatohepatitis model [84]. In a study investigating the effectiveness of HTyr in the renal hypoxia model created by exposing the human proximal tubule cell epithelial cell line (HK-2) to $CoCl_2$ for 24 hours, HTyr treatment reduced the formation of both ROS and reactive nitrogen species (RNS), in addition to prevention of GSH depletion in hypoxia-induced cells. It has also been reported that HTyr has antioxidant, anti-inflammatory and anti-fibrotic effects on hypoxic renal cells by reducing the gene expressions of IL-6 and *TGFß1*, which are modulators of inflammation and fibrosis [29]. Autosomal dominant polycystic kidney disease is a genetic disease that destroys the kidney parenchyma with fluid-filled cysts that lead to kidney failure. The pathological feature of this genetic disease is the development of interstitial inflammation and fibrosis with the accumulation of inflammatory cells. In an in vitro autosomal dominant polycystic kidney disease model, phenolic-rich olive leaf extract decreased the gene expressions of PIC, which play a role in cyst formation, and *TGFß*1, which has a fibrotic effect, fibronectin, eCadherin and α-SMA. In a separate series of experiments, it was reported that *TGFß1* stimulation of cells has been reported to produce a fibrotic effect and that olive leaf extract had an anti-fibrotic effect by reducing *TGFβ1* [85].

According to the our results, statistically important differences among the groups for both *TGFβ1* expression and *TGFβ1* concentraion levels were detected. The *TGFβ1* expresssion, *TGFβ1* and MDA concentration levels of NE group were significantly different ($p < 0.001$) from the both of both of NE+EO and control group. There were significant relation between *TGFβ1* levels and all of LW, CIS, BMS, MDA levels, *TGFβ1* expressions, Casp-3⁺ cells and HPS for lung ($p < 0.001$). Also the relation between MDA levels and all of LW, CIS, BMS, *TGFβ1* expressions, Casp-3⁺ cells and HPS for lung were significant for lung ($p < 0.001$).

Although our NE model is perfectly established, it does not fully represent the pathophysiological events that occur in NE in premature neonates because of the physiological and metabolic differences between rat and human neonates is one of the limitations of our study. Another limitation of our study is that the effects of EO, which has lower total phenolic content, or individual olive oil phenols to determine the effective components were not evaluated. Additionally other limitations of our study are, while the acute effects of EO were determined in our experimental model coversly its' long-term effects were not evaluated, and we evaluated a limited number of parameters related to inflammation and oxidative stress.

## 5. Conclusions

Our findings emphasize that *TGFβ1* has an important role in NE-associated lung injury and can be used as a biomarker to indicate lung injury. In addition, when the protective and therapeutic effects of EO on NE-associated lung injury in newborn rats are evaluated as a treatment

strategy in high-risk newborns, our study provides a promising basis for future clinical studies. We think that future clinical studies including different dose and application time of EO for optimizing EO dosage and duration may provide a new approach to prevent NE-associated lung injury by demonstrating that olive oil may be a safe and effective therapeutic agent could be translate into clinical applications of of high-risk newborns.

## Supporting information

**S1 Fig. Sections from the lung of the control group, Hematoxylin and Eosin staining. Bar: 500 μm.**
(TIF)

**S2 Fig. Sections from the lung of the NE group, Hematoxylin and Eosin staining. Bar: 500 μm.**
(TIF)

**S3 Fig. Sections from the lung of the Treatment group, Hematoxylin and Eosin staining. Bar: 500 μm.**
(TIF)

**S4 Fig. Sections from the lung of the Negative control Caspas-3 staining. Bar: 500 μm.**
(TIF)

**S5 Fig. Sections from the lung of the Control group, Caspas-3 staining. Bar: 500 μm.**
(TIF)

**S6 Fig. Sections from the lung of the NE group, Caspas-3 staining. Bar: 500 μm.**
(TIF)

**S7 Fig. Sections from the lung of the Treatment group, Caspas-3 staining. Bar: 500 μm.**
(TIF)

**S1 Dataset. All data used in the article.**
(XLSX)

## Author contributions

**Conceptualization:** Mustafa Tuşat, Recep Eröz.

**Data curation:** Mustafa Tuşat.

**Formal analysis:** Recep Eröz, Mehmet Semih Demirtaş, Osman Okan Özocak.

**Funding acquisition:** Mustafa Tuşat.

**Investigation:** Mustafa Tuşat.

**Methodology:** Recep Eröz, Ferhan Bölükbaş, Erkan Özkan.

**Project administration:** Mustafa Tuşat.

**Resources:** Mustafa Tuşat.

**Software:** Mehmet Semih Demirtaş, Osman Okan Özocak.

**Supervision:** Recep Eröz.

**Validation:** Mustafa Tuşat, Ferhan Bölükbaş, Erkan Özkan.

**Visualization:** Recep Eröz, Hüseyin Erdal.

**Writing – original draft:** Mustafa Tuşat, Recep Eröz.

**Writing – review & editing:** Mustafa Tuşat, Recep Eröz, Mehmet Semih Demirtaş, Hüseyin Erdal, Osman Okan Özocak.

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
