## [Decision Letter · Decision Letter 0]

25 Oct 2024

PONE-D-24-41150Extra virgin olive oil mitigates lung injury in necrotizing enterocolitis: effects on TGFβ1, Caspase-3, and MDA in a neonatal rat modelPLOS ONE

Dear Dr. tuşat,

Thank you for submitting your manuscript to PLOS ONE. After careful consideration, we feel that it has merit but does not fully meet PLOS ONE’s publication criteria as it currently stands. Therefore, we invite you to submit a revised version of the manuscript that addresses the points raised during the review process.

We look forward to receiving your revised manuscript.

Kind regards,

Rami Salim Najjar, Ph.D.

Academic Editor

PLOS ONE

Journal Requirements:

1. When submitting your revision, we need you to address these additional requirements. Please ensure that your manuscript meets PLOS ONE's style requirements, including those for file naming. The PLOS ONE style templates can be found at https://journals.plos.org/plosone/s/file?id=wjVg/PLOSOne_formatting_sample_main_body.pdf and https://journals.plos.org/plosone/s/file?id=ba62/PLOSOne_formatting_sample_title_authors_affiliations.pdf 2. In the online submission form, you indicated that [The data underlying the results presented in this study can be obtained from the corresponding author].  All PLOS journals now require all data underlying the findings described in their manuscript to be freely available to other researchers, either 1. In a public repository, 2. Within the manuscript itself, or 3. Uploaded as supplementary information.This policy applies to all data except where public deposition would breach compliance with the protocol approved by your research ethics board. If your data cannot be made publicly available for ethical or legal reasons (e.g., public availability would compromise patient privacy), please explain your reasons on resubmission and your exemption request will be escalated for approval.

Additional Editor Comments:

Authors must state how tissue was homogenized and processed for ELISA, including how samples were equally loaded and whether inhibitors were added to the homogenization buffer.

The arrows in figure 2 panels D-G do not show caspase staining and appear arbitrary.

Authors should show entire lung sections with XY stitching along with these zoomed in images to provide confirmation of patterns.

Please disregard Reviewer 2's request for spelled out abbreviations of common terms, however, protein names should be spelled out upon first mention.

Reviewers' comments:

Reviewer's Responses to Questions

**Comments to the Author**

1. Is the manuscript technically sound, and do the data support the conclusions?

Reviewer #1: Yes

Reviewer #2: Yes

Reviewer #3: Partly

2. Has the statistical analysis been performed appropriately and rigorously? 

Reviewer #1: N/A

Reviewer #2: Yes

Reviewer #3: No

3. Have the authors made all data underlying the findings in their manuscript fully available?

Reviewer #1: Yes

Reviewer #2: Yes

Reviewer #3: Yes

4. Is the manuscript presented in an intelligible fashion and written in standard English?

Reviewer #1: Yes

Reviewer #2: Yes

Reviewer #3: No

5. Review Comments to the Author

Reviewer #1: Dear Author,

The manuscript addresses the effects of extra virgin olive oil (EO) on lung injury in a neonatal rat model of necrotizing enterocolitis (NE).

You can find my opinions and suggestions in the attached file.

Best ragrds.

Reviewer #2: 1- The molecular mechanism of TGFβ1 and Caspase-3 should be inserted to introduction section

2- The quality of all figures should be increased

3- The forward and reverse primer sequences of TGFβ1 and GAPDH gene should be inserted to article

4- The conclusion section of article is poor, thus this section should be indicated as detailed

5- There are some spelling errors in generally article, thus this errors should be corrected

Reviewer #3: Major revision requirieds. Changes and suggestion PDF file. Major revision requirieds. Changes and suggestion PDF file. Major revision requirieds. Changes and suggestion PDF file. Major revision requirieds. Changes and suggestion PDF file.

6. PLOS authors have the option to publish the peer review history of their article (what does this mean? ). If published, this will include your full peer review and any attached files.

**Do you want your identity to be public for this peer review?** For information about this choice, including consent withdrawal, please see our Privacy Policy .

Reviewer #1: No

Reviewer #2: **Yes: ** Prof. Dr. Abdullah Aslan

Reviewer #3: No

---

## [Author Response · Author response to Decision Letter 1]

3 Jan 2025

Dear Editor

C1: Authors must state how tissue was homogenized and processed for ELISA, including how samples were equally loaded and whether inhibitors were added to the homogenization buffer.

R1: The sentences ‘’ The tissues were rinsed with pre-cooled PBS to completely remove excess blood before homogenization. Then, an equal weight of tissue (~70 mg) from each group was weighed using a precision scale. Following, 1ml of 1X pre-cooled PBS was added on the tissues and homogenizated via bead mill homogenizer (Scientific industries, digital disruptor genie Cat no:si-dd38) at 2000 rpm for 5 minutes. Finally, the supernatant were collected and stored in aliquots at ≤ -20°C for ELISA.’’ were added in the ‘‘Detection of TGFβ1 and MDA levels via Enzyme‐Linked Immunosorbent Assay (ELISA) in lung tissues’’ section. The inhibitors were not added to the homogenization buffer. All process were quickly performed on the two flat ice packs.

C2: The arrows in figure 2 panels D-G do not show caspase staining and appear arbitrary. Authors should show entire lung sections with XY stitching along with these zoomed in images to provide confirmation of patterns.

R2: In its original form, D represented the negative control. Figures was revised.

Figure 3 was added in manuscript.

The single 4x images accompanying the 20x images of Fig 2 and Fig 3 in the article were uploaded as supporting information.

C3: Please disregard Reviewer 2's request for spelled out abbreviations of common terms, however, protein names should be spelled out upon first mention.

R3: We disregarded Reviewer 2's request for spelled out abbreviations of common terms. However, protein names were spelled out upon first mentioned place throughout the text.

Dear Reviewer 1

C1: The introduction provides a good background on NE and lung injury but lacks a detailed explanation of why EO was chosen. The manuscript should further elaborate on the specific phenolic compounds in EO that might influence the markers studied (TGFβ1, MDA, Caspase-3). This would help link EO's known biological effects to the expected outcomes more clearly

R1: In the introduction section, it was stated that a detailed explanation of why EO was chosen. Additionall, the further elaborate on the specific phenolic compounds in EO that might influence the markers studied (TGFβ1, MDA, Caspase-3) were given and marked with yellow.

C2: While the model's basis is well-described, a justification for the number of animals used (n=8 per group) and the sample size calculation is missing. This information is crucial to validate the statistical power of the study.

R2: In our research, the Resource Equality Method, which is a frequently used method, was used to determine the number of subjects to be assigned to groups. This method, developed for research designs that foresee the ANOVA technique, is based on the Degrees of Freedom error (DFerror). According to Festing and MEad, DFerror between 10 and 20 is sufficient for most experiments with quantitative end-points. In other words, it was assumed that a total sample size of 10 or more DFerror was necessary, and a sample size of 20 or less was sufficient in data analysis.

Based on this, the sample size in our study is;

n=10/k+1 < Sample Size < 20/k+1

k= number of independent groups

N=nk

In the article titled “Determination of Sample Size in Animal Experiments with Resource Equation Method and Power Analysis” published in 2023 by Akbulut Ö., who also prepared a detailed article on this subject, the number of animals per group was taken as 8 by utilizing Table 3 regarding the design suitable for our study, which also includes Cohen's effect sizes and statistical powers.

Table 3. Sample Size and Power Values in Completely Randomized Design (x) (xx)

Number of Groups Sample Limits Sample Size Effect Size (Cohen’s f value)

0.10 0.25 0.40

n N Power Values

Purely Random Process 3 minimum 5 15 0.059 0.110 0.214

maximum 8 24 0.066 0.160 0.353

4 minimum 4 16 0.057 0.097 0.183

maximum 6 24 0.062 0.133 0.289

5 minimum 3 15 0.055 0.082 0.142

maximum 5 25 0.060 0.121 0.259

6 minimum 3 18 0.055 0.086 0.152

maximum 4 24 0.058 0.106 0.215

(x): (α=0.05)

(xx): Doğan İ, Doğan N. Deney hayvanı kullanılan çalışmalarda örneklem büyüklüğünün kaynak eşitlik yöntemi ile tahmini. Türkiye Klinikleri Biyoistatistik Dergisi 2020;12(2):211-217

• Akbulut Ö. KSU Medical Journal 2023;18(2): 117-125 10.17517/ksutfd.1123704

• Festing MFW. Reduction of animal use: experimental design and quality of experiments. Laboratory Animals 1994;28:212- 221.

• Mead R. The Design of Experiments. Cambridge, New York: Cambridge University Press 1988.

Thus, the sentences ‘’ Resource Equality Method, which is a frequently used method, based on the Degrees of Freedom error was used to determine the number of subjects to be assigned to groups.’’ were added in the ‘‘Statistical analysis’’ section.

C3: The methods for scoring histopathological damage and evaluating Caspase-3 expression need further detail. For example, it is important to specify how interobserver variability was managed, as scoring can be subjective.

R3: Histopathologic scoring information was added to the article as Table 2.

Caspase-3 methods section was revised

“Staining of tissue samples taken for immunohistochemical procedures was carried out with a procedure based on streptavidin-biotin-peroxidase complex (sABC). For this purpose, 6 µm thick sections taken on poly L-lysine coated slides were dried by keeping them in an oven at 37oC overnight. Subsequently, the sections were deparaffinized and rehydrated and boiled in a microwave oven in citrate buffer solution (pH 6) for 5 minutes for antigen retrieval. To inhibit endogenous peroxidase activity, the sections were kept in 3% hydrogen peroxide solution for 20 minutes and nonspecific binding sites were blocked by incubating in blocking solution (Thermo Fisher Scientific Inc., UK) for 5 minutes. The sections were then incubated with the primary antibody (Caspase-3 (STJ97448, St John's Laboratory, 1:200 dilution) for 1 hour at room temperature, followed by biotinylated goat polyvalent antibody (Thermo Fisher Scientific Inc., UK) for 30 minutes. Sections washed with buffered phosphate saline (PBS, Biotech) were incubated with streptavidin-peroxidase (HRP, Thermo Fisher Scientific Inc., UK) for 30 min at room temperature. While DAB (3-3'-diaminobenzidine, Thermo Fisher Scientific Inc., UK) was used as chromogen, Mayer's hematoxylin solution was preferred for nuclear staining. Negative control preparations were prepared by incubating tissue sections with PBS instead of primary antibody. The stained sections were covered with coverslips using synthetic adhesive (Entellan, Merck) after being dehydrated through a graded alcohol series and passed through xylene’’ were added in the ‘’ Immunohistochemical Evalution of Lung’’ section.

‘’All evaluations were performed by two researchers blinded to the sample identification’’ added in the ‘’ Immunohistochemical Evalution of Lung’’ section.

C4: While the tables are informative, some figures (e.g., histological images) need improving; low resolution, labels are not clear in some instances and white balance is needed. Without these corrections figures would not proper for publication.

R4: The quality of the figures was improved and checked to ensure that they met PLOS requirements by uploading them to https://pacev2.apexcovantage.com, the Preflight Analysis and Conversion Engine (PACE) digital diagnostic tool that helps ensure that figures meet PLOS requirements.

C5: The results show a significant reduction in TGFβ1 and MDA levels in the EO group. While this supports EO’s anti-inflammatory and antioxidant effects, further discussion on the physiological mechanisms and how EO’s components interact with pathways related to TGFβ1 and oxidative stress would enrich the analysis.

R5: The further discussion on the physiological mechanisms and how EO’s components interact with pathways related to TGFβ1 and oxidative stress to enrich the analysis were given and marked with yellow in the ''Discussion'' section.

C6: Caspase-3 activity is used as a marker of apoptosis. While the results indicate reduced expression in the EO-treated group, providing data on other apoptotic markers (e.g., Bax, Bcl2) could offer a more comprehensive view of EO's anti-apoptotic effects.

R6: Unfortunately, we had not enough financial support to providing data on other apoptotic markers (e.g., Bax, Bcl2).

C7: The discussion addresses the findings well but could be strengthened by comparing them with other studies using similar models or interventions (e.g., other antioxidants). Furthermore, the potential limitations of using neonatal rats as a model for human NE should be acknowledged.

R7: The similar models or interventions to strengthen the discussion were given and marked with yellow. Also potential limitations of using neonatal rats as a model for human NE were given and marked with yellow at the end of ''the Discussion'' section.

C8: The conclusion appropriately highlights EO’s therapeutic potential but would benefit from more emphasis on future directions and clinical applicability.

R8: The future directions and clinical applicability of EO’s therapeutic potential were highlighted and marked with yellow in ''Conclusions'' section.

C9: The manuscript uses several abbreviations, such as NE, EO, and TGFβ1, without consistently defining them upon first use. Ensuring all abbreviations are clearly defined will improve readability.

R9: The all abbreviations were clearly defined in the place of first used throughout the text.

C10: There are some grammatical and typographical errors throughout the manuscript (e.g., "oragastric" should be "orogastric"). A thorough proofreading would be beneficial.

R10: The grammatical and typographical errors were corrcected throughout the manuscript.

Dear Reviewer 2

C1: The molecular mechanism of TGFβ1 and Caspase-3 should be inserted to introductionsection.

R1: The molecular mechanism of TGFβ1 and Caspase-3 were added and marked with yellow in ''introduction'' section.

C2: The quality of all figures should be increased.

R2: The quality of the figures was improved and checked to ensure that they met PLOS requirements by uploading them to https://pacev2.apexcovantage.com, the Preflight Analysis and Conversion Engine (PACE) digital diagnostic tool that helps ensure that figures meet PLOS requirements.

C3: The forward and reverse primer sequences of TGFβ1 and GAPDH gene should be inserted to artic.

R3: The forward and reverse primer sequences of TGFβ1 and GAPDH gene were added and marked with yellow.

C4: The conclusion section of article is poor, thus this section should be indicated as detailed.

R4: The conclusion section of article were reconstructed as detailed.

C5: There are some spelling errors in generally article, thus this errors should be corrected.

R5: The grammatical and typographical errors were corrcected throughout the manuscript.

Dear Reviewer 3

C1: Major revision requirieds. Changes and suggestion PDF file

R1: All suggestions in PDF file offered by reviewer 3 were carried out and marked with yellow.

---

## [Decision Letter · Decision Letter 1]

4 Feb 2025

PONE-D-24-41150R1Extra virgin olive oil mitigates lung injury in necrotizing enterocolitis: effects on TGFβ1, Caspase-3, and MDA in a neonatal rat modelPLOS ONE

Dear Dr. tuşat,

Thank you for submitting your manuscript to PLOS ONE. After careful consideration, we feel that it has merit but does not fully meet PLOS ONE’s publication criteria as it currently stands. Therefore, we invite you to submit a revised version of the manuscript that addresses the points raised during the review process.

We look forward to receiving your revised manuscript.

Kind regards,

Rami Salim Najjar, Ph.D.

Academic Editor

PLOS ONE

Journal Requirements:

Reviewers' comments:

Reviewer's Responses to Questions

**Comments to the Author**

1. If the authors have adequately addressed your comments raised in a previous round of review and you feel that this manuscript is now acceptable for publication, you may indicate that here to bypass the “Comments to the Author” section, enter your conflict of interest statement in the “Confidential to Editor” section, and submit your "Accept" recommendation.

Reviewer #1: All comments have been addressed

Reviewer #2: All comments have been addressed

Reviewer #3: All comments have been addressed

2. Is the manuscript technically sound, and do the data support the conclusions?

Reviewer #1: Yes

Reviewer #2: Yes

Reviewer #3: Yes

3. Has the statistical analysis been performed appropriately and rigorously? 

Reviewer #1: N/A

Reviewer #2: Yes

Reviewer #3: Yes

4. Have the authors made all data underlying the findings in their manuscript fully available?

Reviewer #1: Yes

Reviewer #2: Yes

Reviewer #3: Yes

5. Is the manuscript presented in an intelligible fashion and written in standard English?

Reviewer #1: Yes

Reviewer #2: Yes

Reviewer #3: Yes

6. Review Comments to the Author

Reviewer #1: Dear Author,

Thank you for all the corrections have been made through the manuscript.

You can find mey minor suggestions in the attached file.

Best regards.

Reviewer #2: Thank you for making all the corrections I sent you regarding the revision of the article. Additionally, there is no ethical problem in publishing of the article.

Reviewer #3: Accept submission. Accept submission. Accept submission. Accept submission. Accept submission. Accept submission. Accept submission. Accept submission. Accept submission. Accept submission. Accept submission. Accept submission.

7. PLOS authors have the option to publish the peer review history of their article (what does this mean? ). If published, this will include your full peer review and any attached files.

**Do you want your identity to be public for this peer review?** For information about this choice, including consent withdrawal, please see our Privacy Policy .

Reviewer #1: No

Reviewer #2: **Yes: ** Prof. Dr. Abdullah Aslan

Reviewer #3: No

---

## [Author Response · Author response to Decision Letter 2]

19 Feb 2025

necessary revision was made with consideration of the comments of reviewer 1

---

## [Editor Report · Decision Letter 2]

27 Feb 2025

Extra virgin olive oil mitigates lung injury in necrotizing enterocolitis: effects on TGFβ1, Caspase-3, and MDA in a neonatal rat model

PONE-D-24-41150R2

Dear Dr. tuşat,

We’re pleased to inform you that your manuscript has been judged scientifically suitable for publication and will be formally accepted for publication once it meets all outstanding technical requirements.

Kind regards,

Rami Salim Najjar, Ph.D.

Academic Editor

PLOS ONE
---

## [Editor Report · Acceptance letter]

PONE-D-24-41150R2

PLOS ONE

Dear Dr. Tuşat,

I'm pleased to inform you that your manuscript has been deemed suitable for publication in PLOS ONE. Congratulations! Your manuscript is now being handed over to our production team.

Kind regards,

on behalf of

Dr. Rami Salim Najjar

Academic Editor

PLOS ONE